# mTORC1 inhibition in cancer cells protects from glutaminolysis-mediated apoptosis during nutrient limitation

Victor H. Villar[1], Tra Ly Nguyen[1], Vanessa Delcroix[2], Silvia Terés[1], Marion Bouchecareilh[3], Bénédicte Salin[3], Clément Bodineau[1], Pierre Vacher[2], Muriel Priault[3], Pierre Soubeyran[2] & Raúl V. Durán[1]

A master coordinator of cell growth, mTORC1 is activated by different metabolic inputs, particularly the metabolism of glutamine (glutaminolysis), to control a vast range of cellular processes, including autophagy. As a well-recognized tumour promoter, inhibitors of mTORC1 such as rapamycin have been approved as anti-cancer agents, but their overall outcome in patients is rather poor. Here we show that mTORC1 also presents tumour suppressor features in conditions of nutrient restrictions. Thus, the activation of mTORC1 by glutaminolysis during nutritional imbalance inhibits autophagy and induces apoptosis in cancer cells. Importantly, rapamycin treatment reactivates autophagy and prevents the mTORC1-mediated apoptosis. We also observe that the ability of mTORC1 to activate apoptosis is mediated by the adaptor protein p62. Thus, the mTORC1-mediated upregulation of p62 during nutrient imbalance induces the binding of p62 to caspase 8 and the subsequent activation of the caspase pathway. Our data highlight the role of autophagy as a survival mechanism upon rapamycin treatment.

[1] Institut Européen de Chimie et Biologie, INSERM U1218, Université de Bordeaux, 2 Rue Robert Escarpit, Pessac 33607, France. [2] Institut Bergonié, INSERM U1218, 229 Cours de l'Argonne, Bordeaux 33076, France. [3] Institut de Biochimie et Génétique Cellulaires, CNRS UMR 5095, Université de Bordeaux, 1 Rue Camille Saint-Saëns, Bordeaux 33077, France. Correspondence and requests for materials should be addressed to R.V.D. (email: raul.duran@inserm.fr).

**m**TORC1 (mammalian target of rapamycin complex 1) is a highly conserved serine/threonine kinase complex that integrates several inputs, including amino acid availability, to regulate different cellular processes such as cell growth, anabolism and autophagy[1,2]. mTORC1 pathway is aberrantly activated in 80% of human cancers[3]. Thus, the inhibition of this pathway was considered a relevant approach to treat cancer. However, for still unclear reasons, rapamycin analogues have shown only modest effects in clinical trials[4–6]. Hence, understanding the molecular mechanism by which tumour cells escape from mTORC1 inhibition is a main objective to design new targeted therapies that efficiently eliminate cancer cells. As mTORC1 is strongly regulated by the metabolism of certain amino acids, particularly glutamine, leucine and arginine, there is an intense research nowadays to elucidate how the altered metabolism of amino acids during malignant transformation might play a role in mTORC1 upregulation and in rapamycin treatment resistance.

Glutamine is the most abundant amino acid in the blood and a nitrogen source for cells[7,8]. This amino acid has been described as a crucial nutrient for tumour proliferation, and indeed a vast number of different types of tumour cells consume abnormally high quantities of glutamine and develop glutamine addiction[9–12]. Glutamine is mostly degraded in the cell through glutaminolysis. Glutaminolysis comprises two-step enzymatic reactions, whereby glutamine is first deamidated to glutamate, in a reaction catalysed by glutaminase (GLS), and then glutamate is deaminated to α-ketoglutarate (αKG), in a reaction catalysed by glutamate dehydrogenase. In addition, leucine, another important amino acid from a signalling point of view, activates allosterically glutamate dehydrogenase and promotes the production of glutaminolitic αKG (refs 8,13). Therefore, leucine and glutamine cooperate to produce αKG, an intermediate of the tricarboxylic acid cycle. Besides this anaplerotic role of glutamine, glutaminolysis also activates mTORC1 pathway and inhibits macroautophagy[14]. Macroautophagy (hereafter simply autophagy) is a catabolic process regulated by mTORC1 pathway, through which lysosomal-degradation of cellular components provides cells with recycled nutrients[15–18].

Although it is known that glutaminolysis is a source to replenish tricarboxylic acid cycle and also activates mTORC1, the capacity of glutaminolysis to sustain mTORC1 activation and cell growth in the long term in the absence of other nitrogen sources has not been elucidated. Here we report that, surprisingly, the long-term activation of glutaminolysis in the absence of other amino acids induces the aberrant inhibition of autophagy in an mTORC1-dependent manner. This inhibition of autophagy during amino acid restriction led to apoptotic cell death due to the accumulation of the autophagic protein p62 and the subsequent activation of caspase 8. Of note, the inhibition of mTORC1 restores autophagy and blocks the apoptosis induced by glutaminolysis activation. Our results highlight the tumour suppressor features of mTORC1 during nutrient restriction and provide with an alternative explanation for the poor outcome obtained using mTORC1 inhibitors as an anticancer therapy.

## Results

**Long-term glutaminolysis decreased cell viability.** As we have previously shown that short-term glutaminolysis (15–60 min) is sufficient and necessary to activate mTORC1 and to sustain cell growth (ref. 14), we first explored the capacity of glutaminolysis to serve as a metabolic fuel during amino acid starvation at long term in cancer cells. For the long-term activation of glutaminolysis, we added glutamine (the source of glutaminolysis) and leucine (the allosteric activator of glutaminolysis) to otherwise amino

acid-starved cells as previously described[14], and the cells were incubated in these conditions during 24–72 h. As previously observed, the incubation of a panel of different cancer cell lines, including U2OS, A549 and JURKAT, in the absence of all amino acids arrested cell proliferation, but it did not affect cell viability significantly (Fig. 1a,b and Supplementary Fig. 1A). Strikingly, the activation of glutaminolysis by adding leucine and glutamine (LQ treatment) caused a strong decrease in the number of cells incubated in these conditions (Fig. 1a,b and Supplementary Fig. 1B). Similar results were obtained in HEK293 cells (Fig. 1a,b). To confirm whether this decrease in the number of cells was related to an increase in cell death or a decrease in cell proliferation, we measured the percentage of cell death using the trypan blue exclusion assay, and we determined cell viability using a clonogenic assay. We observed that glutaminolysis activation using LQ treatment increased the percentage of cell death in all the tested cell lines (Fig. 1c and Supplementary Fig. 1C). In addition, LQ treatment during amino acid restriction strongly reduced the number of colonies formed in a clonogenic assay (Fig. 1d,e and Supplementary Fig. 1D). Further confirming that glutaminolysis was responsible for cell death induction upon LQ treatment, we used a cell-permeable derivative of αKG, dimethyl-α-ketoglutarate (DMKG), and we observed that the addition of DMKG to amino acid-starved cells induced cell death to a similar extent than LQ treatment (Fig. 1a–c). As αKG is the final product of glutaminolysis, and we previously showed that LQ treatment efficiently increases the intracellular levels of αKG (ref. 14), this result suggested that the ability of LQ treatment to induce cell death correlates with the activation of glutaminolysis and αKG production. To finally confirm the active role of glutaminolysis in LQ-induced cell death, we inhibited the enzyme GLS1 (both genetically using siRNA and pharmacologically using the inhibitors DON and BPTES), responsible for the first step of glutaminolysis. As shown in Fig. 1f–i and in Supplementary Fig. 1E,F, both the genetic and the pharmacologic inhibition of GLS1 prevented the LQ-induced cell death in U2OS and HEK293 cells, with no effect on the viability of the cells during amino acid starvation. Of note, GLS1 inhibition using DON or GLS1 silencing using siRNA did not prevent the induction of cell death mediated by DMKG treatment (Supplementary Fig. 1G,H), as DMKG bypasses the inhibition of GLS1 to produce αKG. Taken together, these results strongly support the conclusion that the unbalanced production of αKG by glutaminolysis during amino acid restriction decreases cell viability. The specificity of glutaminolysis in the observed cell death was confirmed as the addition of all the amino acids did not decrease cell viability (Supplementary Fig. 1I,J). Thus, glutaminolysis (a well-known pro-proliferative process) causes cell death if it is activated during nutrient restriction, a result that pointed at the importance of nutritional balance in the control of cancer cell viability.

**Unbalanced glutaminolysis induced apoptosis.** We next investigated whether the cell death induced by glutaminolysis was apoptosis. For this purpose, we analysed several apoptotic markers, such as the cleavage of caspases 3, 8 and 9, the cleavage of PARP, and the expression of the pro-apoptotic protein BAX after 72 h of amino acid restriction in several tumour cell lines, by western blot and by immunofluorescence. We also analysed the expression of the anti-apoptotic proteins Bcl-XL and MCL-1. We observed that LQ treatment or DMKG treatment increased the levels of cleaved caspase 3, cleaved PARP and BAX in U2OS (Fig. 2a–c), A549 (Supplementary Fig. 2A), JURKAT (Supplementary Fig. 2B) and HEK293A cells (Supplementary Fig. 2D,F), in a dose-dependent manner (Supplementary Fig. 2C). We also observed an increase in the cleavage of caspase 8, while we

did not detect any changes in cleaved caspase 9 (Fig. 2a). Similarly, we did not see any decrease in the levels of the anti-apoptotic proteins Bcl-XL and MCL-1 (Fig. 2a). In agreement with these results, both LQ and DMKG treatment significantly increased the late apoptotic population compared to amino acid-starved cells as determined by the double positive annexin V/PI staining observed by flow cytometry (Fig. 2d,e). Further confirming that LQ-induced cell death is mediated by an activation of apoptosis, we used zVAD-FMK, a specific inhibitor of caspases. As shown in Fig. 2f,g, the treatment of cells with zVAD-FMK completely abolished the LQ-mediated cell death and activation of the apoptotic markers, supporting that LQ-induced cell death can be explained by an increase in apoptosis.

We next assayed whether αKG was a necessary byproduct for the induction of apoptosis upon LQ treatment. For this purpose, we inhibited glutaminolysis either genetically (siRNA GLS1) and pharmacologically (using DON or BPTES). Following this approach, we confirmed that the silencing of GLS1 or the inhibition of GLS1 drastically reduced the induction of cleaved caspase 3, cleaved PARP and BAX by LQ treatment (Fig. 2h,i and Supplementary Fig. 2E,F). Likewise, the pharmacological inhibition of GLS1 blocked the increase in the population of annexin V/PI-positive cells induced by LQ (Supplementary Fig. 2G,H). In contrast to LQ treatment, and as expected, the addition of all amino acids did not induce apoptosis (Supplementary Fig. 2I). Therefore, we concluded that the

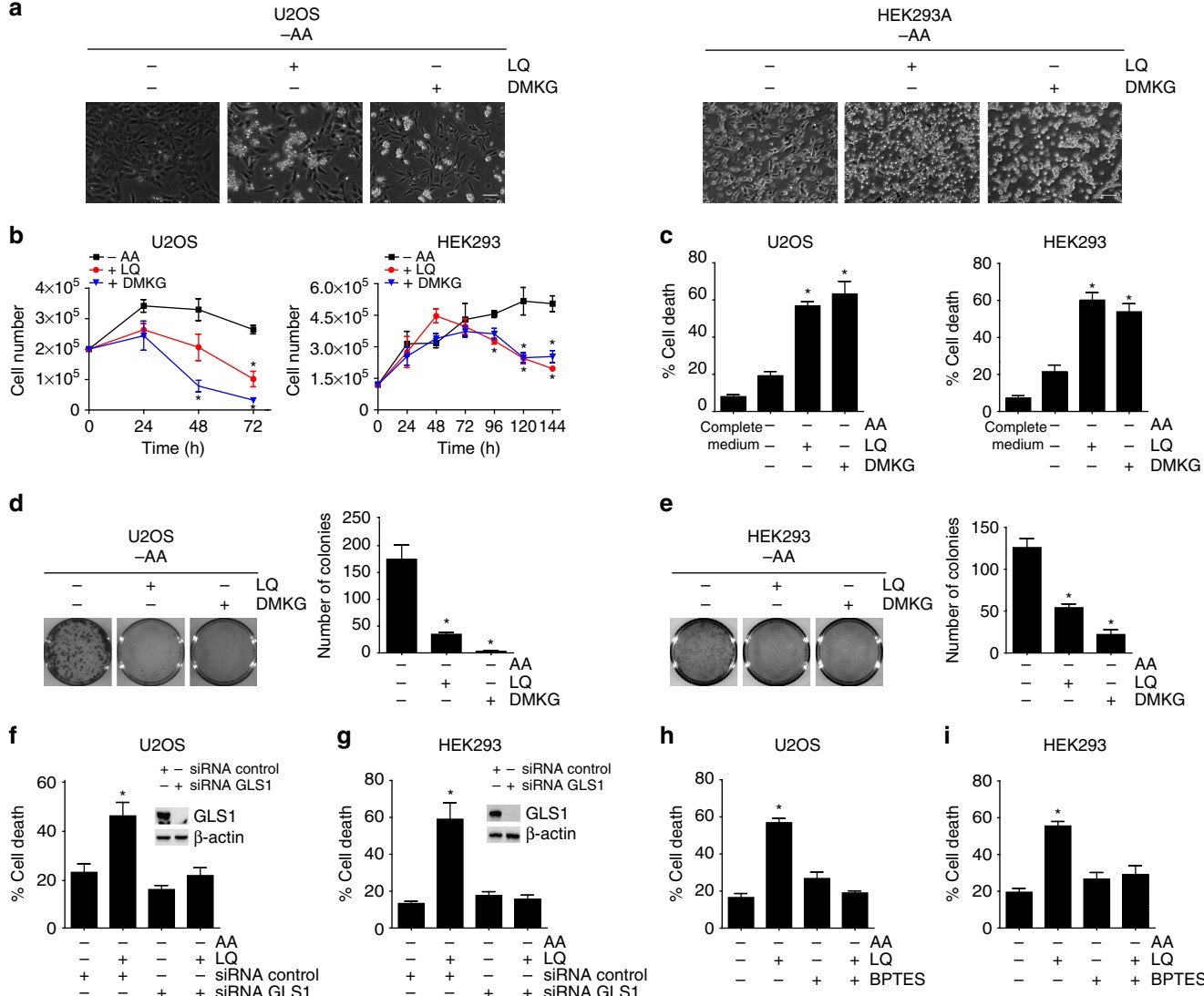

**Figure 1 | Long-term glutaminolysis activation during amino acid restriction decreased cell viability.** (**a**) U2OS (left panel) and HEK923 (right panel) cells were starved for all the amino acids (−AA) in the presence or absence of LQ or DMKG (2 mM) for 72 h (U2OS) or 144 h (HEK293). Representative microscopy images of the cells are shown for the indicated conditions. The scale bar represents 100 μm. (**b**) Proliferation curves for U2OS and HEK293 were determined upon −AA, in the presence or the absence of LQ and DMKG after 24–144 h. (**c**) Percentage of cell death was estimated using trypan blue exclusion assay upon LQ or DMKG treatment after 72 h for U2OS or 144 h for HEK293, as indicated. (**d,e**) A representative image of a clonogenic assay (left panel) and the quantification of the colonies formed in three independent experiments (right panel) are shown for U2OS (**d**) and HEK293 (**e**). (**f,g**) Percentage of cell death was estimated in cells depleted of GLS1 (siRNA GLS1) upon amino acids starvation either in the presence or the absence of LQ after 72 h for U2OS (**f**) and 144 h for HEK293 (**g**). (**h,i**) Percentage of cell death was estimated upon amino acids starvation either in the presence or the absence of LQ and BPTES (30 μM) after 72 h for U2OS and 144 h for HEK293 cells. Graphs show mean values ± s.e.m. ($n = 3$). *$P < 0.05$ (Anova *post hoc* Bonferroni).

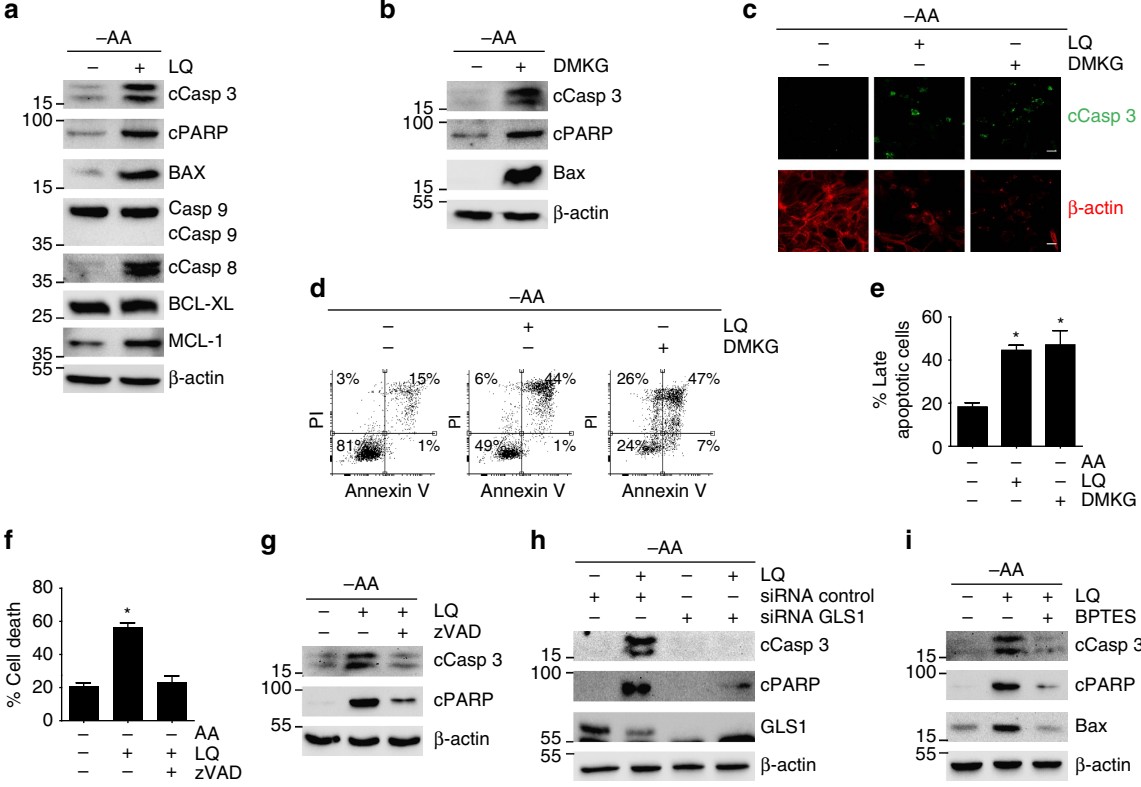

**Figure 2 | Glutaminolysis activation during amino acid restriction induced apoptosis.** (**a,b**) U2OS cells were starved for amino acid either in the presence or the absence of LQ (**a**) or DMKG (2 mM) (**b**). The level of the pro-apoptotic proteins (caspase 3, PARP, BAX, Caspase 8 and caspase 9) and anti-apoptotic member of the Bcl-2 family (Bcl-XL and MCL-1) were determined by western blot for U2OS cells treated as indicated. The scale bar represents 20 μm. (**c**) Immunofluorescence analysis of cleaved caspase 3 and actin filaments are shown for LQ-treated and DMKG-treated cells upon amino acid starvation after 72 h in U2OS cells. (**d**) Flow cytometry analysis of annexin V/PI staining of U2OS cells treated with LQ or DMKG as indicated. (**e**) Quantification of late apoptosis (annexin V/PI-positive cells) for the indicated conditions in U2OS cells. (**f,g**) Effect of the inhibition of apoptosis using zVAD-FMK (1 μM) on the percentage of cell death (**f**) and apoptotic markers (**g**) in LQ-treated U2OS cells. (**h,i**) Western blot analysis of apoptotic markers upon GLS1 silencing using siRNA (**h**) or upon GLS inhibition using BPTES (**i**) in LQ-treated U2OS cells. Graphs show mean values ± s.e.m. ($n = 3$). *$P < 0.05$ (Anova *post hoc* Bonferroni).

increase in glutaminolytic αKG production during amino acids restriction induced apoptotic cell death.

In order to gain some insights in the mechanism of apoptosis induction mediated by LQ, we investigated if the observed increased levels of the pro-apoptotic protein BAX played a mechanistic role in the induction of apoptosis. Indeed, the downregulation of BAX using siRNA reduced cell death, caspase 3/8 cleavage, and PARP cleavage in LQ-treated cells (Supplementary Fig. 2J,K), suggesting that BAX upregulation is necessary for the activation of apoptosis by LQ treatment. This was a rather surprising result, as we previously observed an activation of caspase 8, and not caspase 9 (Fig. 2a), as canonically described for BAX (ref. 19). Indeed, the increased protein levels of BAX observed in LQ treated cells did not correlate with an increase in BAX transcription, as no significant differences in BAX mRNA levels were observed upon LQ addition (Supplementary Fig. 2L). Accordingly, both the levels and activity of p53 (as determined by p21 as a readout), a major transcriptional regulator of BAX (ref. 20), were decreased upon glutaminolysis activation (Supplementary Fig. 2M), further discarding an increase in the transcription of BAX in LQ-treated cells. Finally, we also investigated if LQ-treated cells showed a BAX-mediated induction of the intrinsic pathway of apoptosis that promotes the release of cytochrome c from the mitochondria, which in turn activates the caspase 9. As shown in Supplementary Fig. 2N, LQ treatment did not induce the release of cytochrome c from the mitochondria, in agreement with the lack of caspase

9 cleavage previously observed. Thus, we concluded that BAX played a non-canonical mechanistic role in LQ-induced apoptosis.

**mTORC1 inhibition prevented glutaminolysis induced apoptosis.** We have shown previously that short-term (15 min) glutaminolysis is sufficient and necessary to activate mTORC1 pathway[14]. However, whether glutaminolysis is sufficient to activate mTORC1 at long term (72 h) was not clear. Hence, we investigated if the induction of apoptosis mediated by the activation of glutaminolysis in the absence of other amino acids correlated with the activation of mTORC1. Indeed, the activation of glutaminolysis adding LQ to amino acid-starved cells, or the treatment with DMKG, increased the phosphorylation S6K (Thre389), S6 (Ser235/236) and 4EBP1 (Ser37/46), all of them downstream targets of mTORC1, after 72 h (Supplementary Fig. 2A–I). In contrast, LQ treatment did not affect the activation of mTORC2, as the phosphorylation of its downstream target AKT at Ser473 (refs 2,21) was not affected (Fig. 3d). These results confirmed that long-term activation of glutaminolysis adding glutamine and leucine, even in the absence of other amino acids, is sufficient to activate mTORC1, but has not effect towards mTORC2, upon amino acid starvation. In addition, the pharmacological inhibition of GLS (using DON or BPTES) abrogated the activation of mTORC1 induced by LQ treatment, supporting a mechanistic role of glutaminolytic αKG

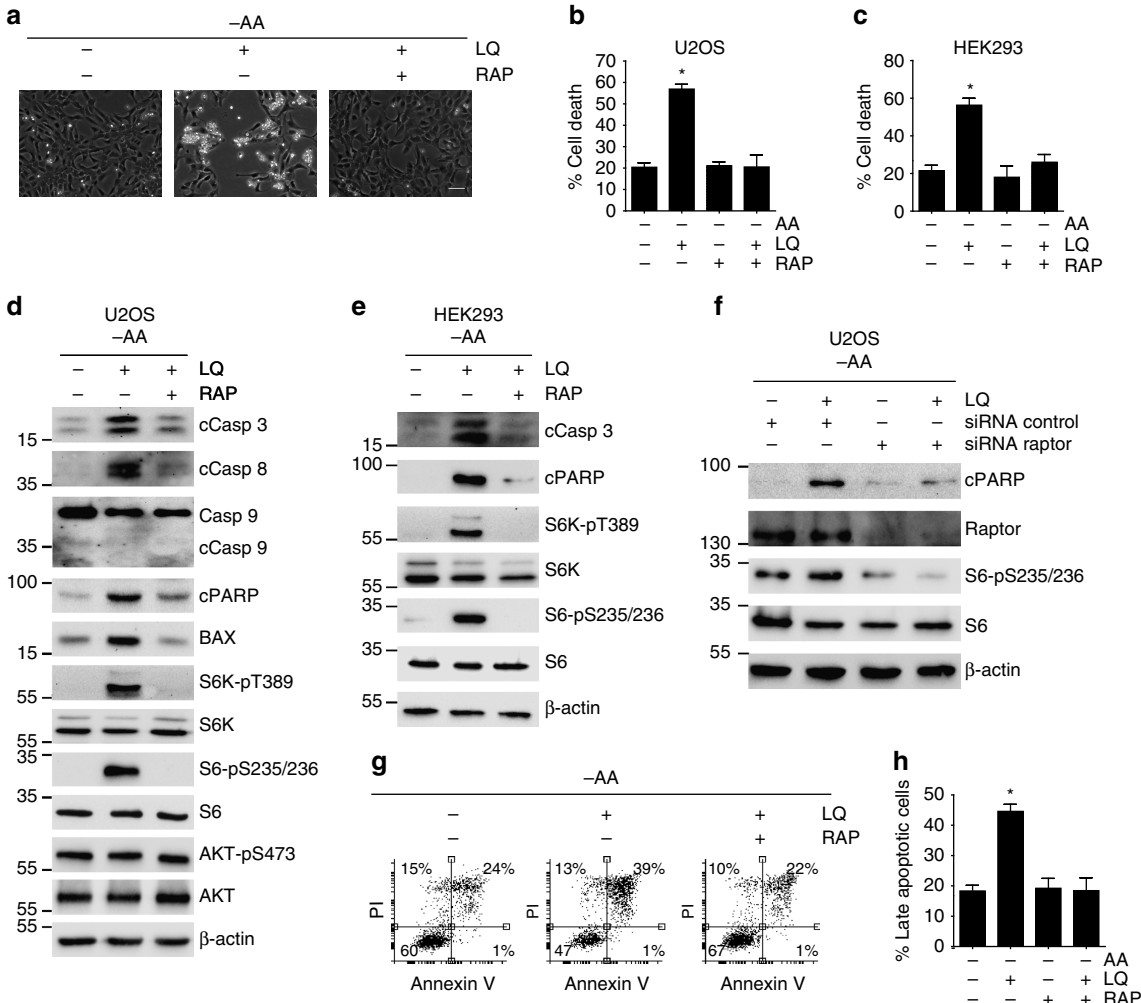

**Figure 3 | mTORC1 inhibition prevented the glutaminolysis induced apoptosis.** (**a**) Representative microscopy image of U2OS cells upon LQ treatment either in the presence or the absence of rapamycin after 72 h. The scale bar represents 100 μm. (**b,c**) Percentage of cell death as estimated using trypan blue exclusion assay is U2OS cells (**b**) or HEK293 cells (**c**) upon LQ treatment either in the presence or the absence of rapamycin for 72 h (U2OS) or 144 h (HEK293). (**d,e**) Western blot analysis of apoptotic markers and mTORC1 downstream targets upon rapamycin (RAP) addition in LQ-treated U2OS cells (**d**) and HEK293 cells (**e**). (**f**) Western blot analysis of apoptotic markers and mTORC1 downstream targets upon the silencing of Raptor using siRNA (thus inhibiting mTORC1 activity) in LQ-treated U2OS cells. (**g**) Flow cytometry analysis of annexin V/PI staining of U2OS cells treated with LQ and rapamycin as indicated. (**h**) Quantification of late apoptosis (annexin V/PI-positive cells) for the U2OS cells treated as in **g**, as indicated. Graphs show mean values ± s.e.m. ($n = 3$). *$P < 0.05$ (Anova *post hoc* Bonferroni).

levels in the activation of mTORC1 at long term, as previously demonstrated for short-term glutaminolysis[14] (Supplementary Fig. 2E–H).

The translocation of mTORC1 to the surface of the lysosome is a crucial step for its activation[22], and indeed, we previously showed that short-term glutaminolysis was sufficient to induce the lysosomal translocation of mTORC1 (ref. 14). Hence, we evaluated whether long-term glutaminolysis was also sufficient to sustain the localization of mTORC1 at the surface of the lysosome. For this purpose, we determined the colocalization between mTOR and CD63, a late endosomal and lysosomal marker, by confocal microscopy. As shown in Supplementary Fig. 3C, mTOR showed a disperse distribution throughout the cytoplasm when cancer cells were incubated in the absence of amino acids. In contrast, the activation of glutaminolysis by either LQ treatment or DMKG treatment in amino acid-restricted cells was sufficient to sustain the co-localization of mTOR and CD63 even after 72 h of treatment, again confirming that long-term glutaminolysis is sufficient to induce the lysosomal localization of mTORC1, prior to its activation. Furthermore, as mTORC1

activity regulates cell size[23], we also confirmed that the activation of mTORC1 by long-term glutaminolysis increased cell size. As shown in Supplementary Fig. 3D, while the withdrawal of amino acids decreased cell size, the activation of glutaminolysis (LQ treatment) maintained the size of the cells to a similar level than cells grown in a complete medium. As expected, the inhibition of mTORC1 using rapamycin in LQ-treated cells decreased the cell size to a similar extent than amino acids starvation. Altogether, these results strongly support that glutaminolysis is sufficient to maintain mTORC1 active upon amino acid deprivation at long term (72 h). This is an abnormal activation of mTORC1, as the mTORC1-dependent activation of cell growth does not seem viable for a prolonged time in conditions of amino acid restriction. Indeed, it is already known that the hyperactivation of mTORC1 in TSC2 −/− MEFs leads to cell death upon nutrient deprivation[24–26]. Therefore, we hypothesized that this unbalanced activation of mTORC1 may be mechanistically linked to the glutaminolysis-mediated apoptosis.

To evaluate the mechanistic link between the activation of mTORC1 and glutaminolysis-induced apoptosis, we followed

both a genetic and pharmacological approach to inhibit mTORC1 upon LQ treatment to determine whether the induction of apoptosis was affected. First, we observed that mTORC1 inhibition using rapamycin efficiently prevented the increase in cell death mediated by glutaminolysis in U2OS, and HEK293 cells (Fig. 3a–c). We next confirmed that the activation of glutaminolysis adding LQ to amino acid-starved cells induced the concomitant activation of mTORC1, as determined by S6K phosphorylation and S6 phosphorylation, and the activation of apoptosis, as determined by the cleavage of caspase 3, caspase 8 and PARP, in several cellular models (Fig. 3d,e and Supplementary Fig. 3E). Very importantly, the efficient pharmacological inhibition of mTORC1 using rapamycin (assessed by the reduction in the phosphorylation of mTORC1 downstream targets S6K, S6 and 4EBP1) completely prevented the activation of caspase pathway by glutaminolysis, as rapamycin treatment in LQ-induced cells was sufficient to drastically reduce the cleavage of caspase 3, caspase 8 and PARP, and to prevent the upregulation of BAX (Fig. 3d,e and Supplementary Fig. 3E). Similar results were obtained with a different inhibitor of mTORC1 (PP242, which is a dual inhibitor of both mTORC1 and mTORC2), and with the genetic inhibition of mTORC1 using an siRNA that efficiently silenced Raptor, a mTORC1-specific component[27,28] (Fig. 3f and Supplementary Fig. 3F). In contrast, the genetic inhibition of mTORC2 using an siRNA against Rictor did not block apoptosis induction (Supplementary Fig. 3G). Furthermore, mTORC1 inhibition using rapamycin abrogated the increase in the late apoptotic population mediated by LQ treatment. Thus, as shown in Fig. 3g,h, although rapamycin treatment did not affect the percentage of annexin V/PI-positive population in amino acid-starved cells, it prevented the increase in the percentage of annexin V/PI-positive cells induced by LQ.

The results presented above strongly suggest that glutaminolysis-induced apoptosis is mediated by the aberrant activation of mTORC1 during amino acid limitation. This very important observation suggests that mTORC1, besides its well-known function as a tumour promoter, also exhibits tumour suppressor features during nutritional limitation, a conclusion with important consequences in terms of targeting mTOR as anticancer-therapy. Of note, the capacity of unbalanced glutaminolysis to induce cell death was also evident in a genetic background showing mTORC1 upregulation, such as the case of TSC2 − / − MEFs. Indeed, we observed that LQ-treatment-induced caspase 3 cleavage in TSC2 − / − MEFs, and rapamycin treatment efficiently abrogated this effect (Supplementary Fig. 3H), underscoring the physiological relevance of our finding in a genetic context of mTORC1 overactivation.

**UPR did not participate in glutaminolysis mediated apoptosis.** In order to understand the cellular mechanism by which the unbalanced activation of mTORC1 induced apoptosis during nutrient limitation, we first investigated the potential role of endoplasmic reticulum (ER) stress and unfolded protein response (UPR) (refs 24,29,30) upon glutaminolysis activation. As shown in Supplementary Fig. 3I, we did not observe any change in the levels of Bip/GRP78, elF2α-pS52, HErpud1 or PDI upon LQ treatment as determined by western blot, all of them being markers of ER stress and UPR. Similarly, the inhibition of mTORC1 using rapamycin in these conditions did not affect the levels of those proteins. This result suggests that the activation of glutaminolysis and mTORC1 during amino acid restriction did not induce an ER stress, neither activated UPR by the time at which apoptosis was already activated. This result does not support an active role of UPR and ER stress in the mechanism of apoptosis activation mediated by mTORC1.

As we observed that LQ-treatment induced the activation of caspase 8, we investigated if the extrinsic pathway of apoptosis (activated by apoptosis-inducing ligands such as FasL, TRAIL or TNFα, and leading to the activation of caspase 8 (ref. 31)) was involved in LQ-mediated apoptosis, and whether rapamycin treatment had an effect on the capacity of the extrinsic pathway to induce apoptosis. However, we did not observe any change in the levels of FasL, TRAIL or TNFα upon LQ treatment either in the presence or the absence of rapamycin (indeed, no detectable levels of FasL or TNFα were appreciated in WB). Similarly, we did not observe any change in the levels of the death receptor Fas (Supplementary Fig. 3J). This result discards that mTORC1 inhibition restricted translation of TNFα, FasL or TRAIL as a mechanism to explain cell death via caspase 8. Further sustaining this conclusion, rapamycin treatment did not block the capacity of FasL or TRIAL to induce apoptotic cell death (Supplementary Fig. 3K–N), discarding that the protective effect of rapamycin treatment involved the modulation of ligand-induced apoptosis.

**Autophagy inhibition associated to unbalanced glutaminolysis.** The activation of mTORC1 is known to inhibit the initiation of autophagy[32–36]. Autophagy is a degradative process that maintains metabolism and survival in condition of nutrient limitation[8,16]. Previously we have reported that the short-term activation of glutaminolysis inhibits autophagy in an mTORC1-dependent manner[14]. Now, we wanted to explore whether the activation of apoptosis observed upon the unbalanced activation of glutaminolysis/mTORC1 during nutrient restriction correlated with an inhibition of autophagy. First, and as previously shown, we observed that the inhibition of mTORC1 induced by long-term amino acids withdrawal (8–72 h) led to the induction of autophagy. As shown in Fig. 4a,b and Supplementary Fig. 4A,B, U2OS cells stably expressing a GFP-LC3 construct displayed an increase in the number of GFP-LC3 aggregates after 8–72 h of amino acids starvation with respect to cells incubated in the presence of amino acids, clearly suggesting an increase in autophagosome formation[37]. In contrast, LQ and αKG treatment strongly decreased the number of GFP-LC3 aggregates in U2OS cells, suggesting that glutaminolysis is sufficient to inactivate autophagy during amino acids restriction. In agreement with this conclusion, LQ and αKG treatments were also sufficient to increase the expression of p62 and to decrease the formation of LC3II (Fig. 4b,c), indicating a decrease in autophagy[37]. We also evaluated the autophagic flux using chloroquine (CQ) to trap the formation of autophagosomes. Whereas the addition of CQ to amino acid-starved cells increased GFP-LC3 punctate and the levels of LC3II, LQ or αKG treatments were able to reduce GFP-LC3 punctate, to reduce the levels of LC3II and to increase the levels of p62 even in the presence of CQ (Supplementary Fig. 4A–D), suggesting that indeed glutaminolysis reduced the autophagic flux. Finally, we also determined autophagy activation through the analysis of autophagy-related vesicles by transmission electron microscopy. While cells exposed to amino acid starvation for 72 h displayed an increase in the autophagy-related vesicles, the activation of glutaminolysis upon LQ treatment blocked the formation of those vesicles (Fig. 4d), again suggesting an inhibition of autophagy. Thus, the apoptotic cell death induced by the activation of glutaminolysis in the absence of amino acids correlated with an inhibition of autophagy.

To elucidate the role of mTORC1 on the blockage of autophagy by glutaminolysis, we inhibited mTORC1 pathway using rapamycin. As shown in Fig. 4a,b and Supplementary Fig. 4A–C, the inhibition of mTORC1 using rapamycin prevented the

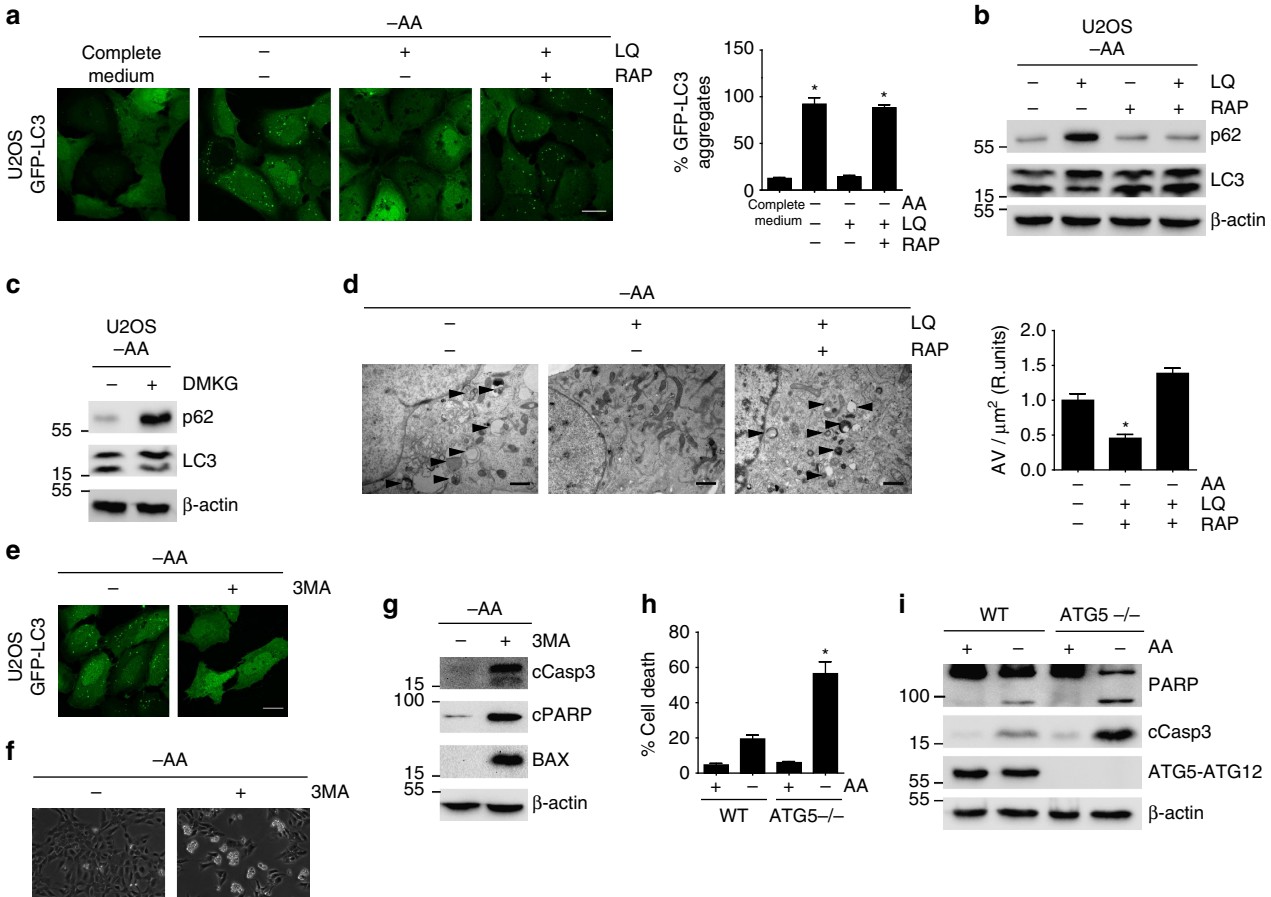

**Figure 4 | Glutaminolysis activated cells showed an mTORC1 dependent inhibition of autophagy during amino acid restriction.** (**a**) GFP-LC3 expressing U2OS cells were starved for amino acids in the presence or absence of LQ and RAP for 72 h as indicated. Autophagosome formation upon GFP-LC3 aggregation was determined (left panel) and quantified (right panel) using confocal microscopy. The scale bar represents 20 μm. (**b,c**) Western blot analysis of U2OS cells treated with LQ, DMKG and RAP as indicated to determine the levels of p62 and LC3II after 72 h. (**d**) Transmission electron microscopy (TEM) images of U2OS cells starved for amino acid in the presence or absence of LQ and RAP after 72 h. The number of autophagy-related vesicles per μm$^2$ was quantified for each indicated condition. The scale bar represents 1 μm. (**e**) GFP-LC3 expressing U2OS cells were starved for amino acids in the presence or absence of 3MA (5 mM) during 72 h. Autophagosome formation upon GFP-LC3 aggregation was determined using confocal microscopy. The scale bar represents 20 μm. (**f,g**) Representative microscopy image (**f**) and western blot analysis of apoptotic markers (**g**) in U2OS cells upon 3MA treatment during 72 h as indicated. The scale bar represents 100 μm. (**h,i**) WT and ATG5 − / − MEFs were incubated either in the presence ( + AA) or the absence ( − AA) of amino acids ( + AA) for 24 h. Cell viability using trypan blue exclusion assay (**h**) and western blot analysis of apoptotic markers (**i**) are shown. Graphs show mean values ± s.e.m. (n = 3). *P < 0.05 (Anova *post hoc* Bonferroni).

LQ-mediated inhibition of autophagy flux, as determined by the increasing number of GFP-LC3 aggregates, the decreasing levels of p62 protein and the increasing formation of LC3II. Furthermore, the inhibition of mTORC1 using rapamycin also blocked the capacity of glutaminolysis to reduce the number of autophagy-related vesicles (Fig. 4d). Similar results were observed upon the genetic inhibition of mTORC1 using an siRNA that efficiently silenced Raptor protein. As shown in Supplementary Fig. 4E,F, Raptor (but not Rictor) depletion blocked the inhibition of autophagy by glutaminolysis, as assessed by the increasing levels of the GFP-LC3 aggregates and the decreasing levels of p62. Finally, to confirm that the glutaminolitic flux inhibits autophagy, we inhibited glutaminolysis using either DON or BPTES, and using an siRNA that efficiently silenced GLS1. In all the cases, the inhibition of GLS prevented the inhibition of autophagy by LQ treatment, assessed by the decrease of GFP-LC3 aggregates, the decrease in p62 levels and the increase in LC3II (Supplementary Fig. 4G–I). All those results confirmed that the capacity of long-term LQ treatment to inhibit autophagy required glutaminolysis and the activation of mTORC1.

Next, we wanted to investigate whether the inhibition of autophagy mediated by the activation of glutaminolysis/mTORC1 plays a mechanistic role in the induction of apoptosis during amino acid restriction. For this purpose, we first investigated whether autophagy is necessary to sustain cell viability upon nutrient restriction. Hence, we inhibited autophagy pharmacologically using 3-methyladenine (3MA) during amino acid withdrawal. 3MA treatment efficiently inhibited autophagy in amino acid-starved cells, decreasing the formation of GFP-LC3 aggregates (Fig. 4e), and induced apoptotic cell death as determined by the levels of cleaved caspase 3, cleaved PARP and BAX (Fig. 4f,g). Similarly, the inhibition of autophagy using CQ, also induced apoptotic cell death in amino acid-restricted cells (Supplementary Fig. 4J,K). These results confirmed that autophagy is necessary to sustain cell viability in conditions of amino acid restriction. We confirmed the results obtained with 3MA and CQ using ATG5 − / − MEFS and ATG5 knock down in U2OS cells. The ablation/downregulation of ATG5 prevented the capacity of the cells to survive upon amino acid withdrawal (Fig. 4h,i and Supplementary Fig. 5C).

This conclusion led us to investigate whether autophagy inhibition is the mechanistic link between the activation of glutaminolysis/mTORC1 during amino acid restriction and apoptosis.

**Autophagy mediated cell survival upon rapamycin treatment.** To elucidate if autophagy inhibition is the mechanistic link between the activation of glutaminolysis/mTORC1 and apoptosis during amino acid restriction, we tested whether the inhibition of autophagy prevents the rapamycin-dependent restoration of cell viability upon LQ treatment. The treatment of cells with 3MA was sufficient to inhibit autophagy in rapamycin-treated cells (Fig. 5a). Importantly, the inhibition of autophagy using 3MA or CQ also prevented the rapamycin-mediated inhibition of apoptosis in glutaminolysis-activated cells, as determined by trypan blue exclusion assay (Fig. 5b,c) and apoptotic markers such as cleaved caspase 3, cleaved PARP and BAX (Fig. 5d and Supplementary Fig. 5A), while no reactivation of mTORC1 was detected. Similarly, 3MA treatment completely abolished the capacity of rapamycin to restore colony number in a clonogenic assay (Fig. 5e,f). These results were confirmed using ATG5 knock down in U2OS cells and ATG5 − / − MEFs, in which the impairment of autophagy completely abolished the

capacity of rapamycin to promote cell survival during amino acids starvation (Fig. 5g,h and Supplementary Fig. 5C–E). As expected, in ATG5 − / − MEFs, neither LQ treatment nor rapamycin modulated autophagy, as assessed by the levels of p62, which remained high due to the inactivation of autophagy (Supplementary Fig. 5B). All these results confirmed that the ability of rapamycin to promote apoptosis resistance necessarily requires autophagy re-activation, confirming that the mechanism of glutaminolysis-induced apoptosis is mediated by autophagy inhibition.

**p62 mediated caspase 8 activation and apoptosis induction.** While the previous results strongly suggested that mTORC1 inhibition prevented apoptotic cell death in an autophagy-dependent manner, we wanted to elucidate the precise molecular mechanism of apoptosis induction resulting from the unbalanced activation of mTORC1. The contribution of p62 to the activation of caspase 8, caspase 3 and apoptosis under certain stress conditions has been described previously[38–40]. Indeed, the interaction between p62 and caspase 8 has been reported to activate caspase pathway and apoptosis[41]. p62 levels are normally downregulated in amino acid-restricted conditions due to the activation of autophagy[42]. However, the activation of glutaminolysis/mTORC1

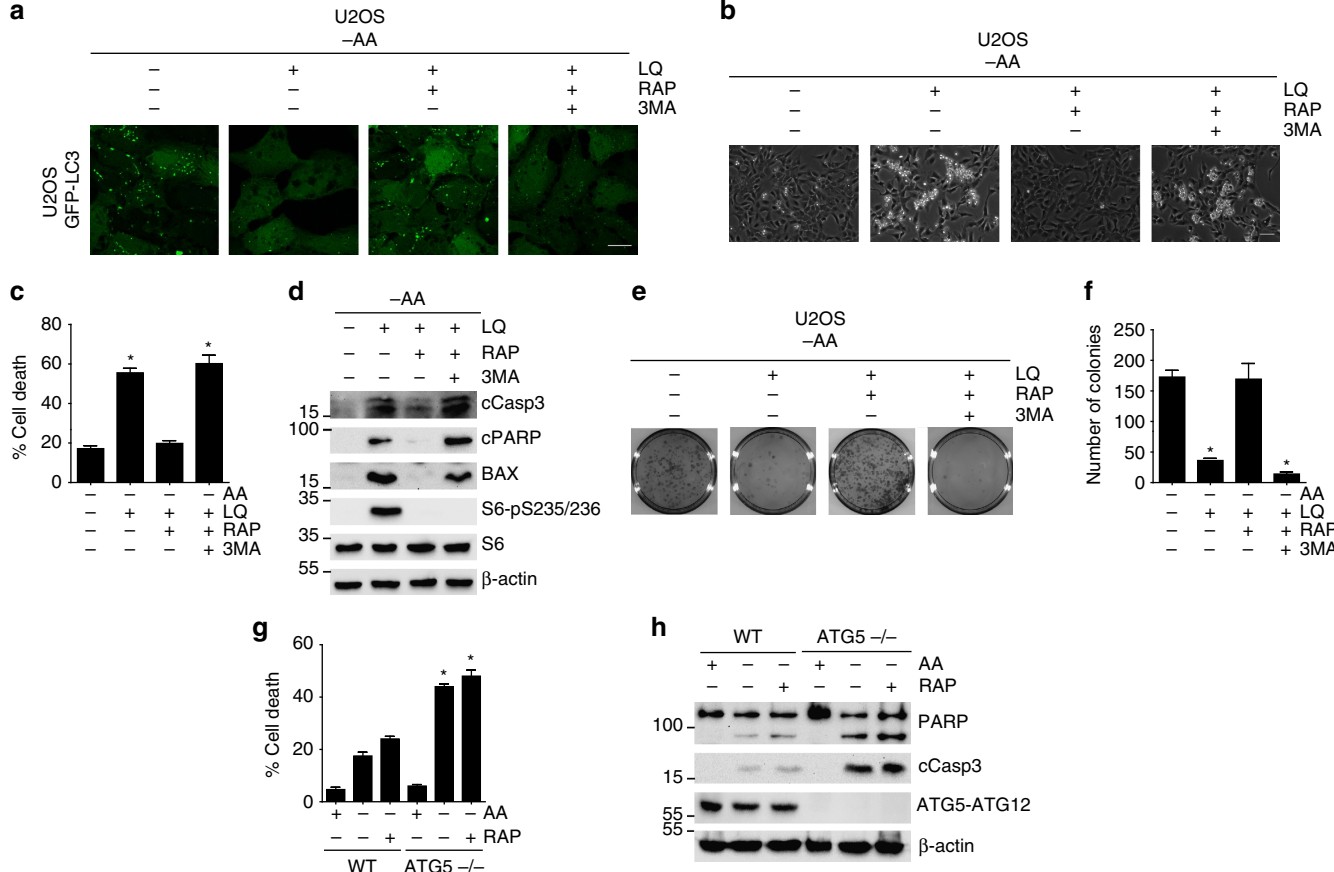

**Figure 5 | Autophagy was necessary for the ability of rapamycin treatment to prevent glutaminolysis and mTORC1 induced apoptosis.** (**a**) GFP-LC3 expressing U2OS cells were starved for amino acids in the presence or absence of LQ, RAP and 3MA for 72 h as indicated. Autophagosome formation upon GFP-LC3 aggregation was determined using confocal microscopy. The scale bar represents 20 μm. (**b**) U2OS cells were starved for all the amino acids in the presence or absence of LQ, RAP and 3MA for 72 h as indicated. A representative microscopy image of the cells for the indicated conditions is shown. The scale bar represents 100 μm. (**c**) Percentage of cell death as estimated using trypan blue exclusion assay in U2OS treated as in **b**. (**d**) Western blot analysis of apoptotic markers and mTORC1 downstream targets was assessed for U2OS cells treated as in **b**. (**e,f**) Clonogenic assay of U2OS cells treated as in **b** (**e**). The number of colonies in three independent experiments was quantified (**f**). (**g,h**) WT and ATG5 − / − MEFs were incubated either in the presence ( + AA) or the absence ( − AA) of amino acids ( + AA) and rapamycin as indicated for 24 h. Cell viability using trypan blue exclusion assay (**g**) and western blot analysis of apoptotic markers (**h**) are shown. Graphs show mean values ± s.e.m. (n = 3). *P < 0.05 (Anova post hoc Bonferroni).

clearly sustained high levels of p62 in amino acid-restrictive conditions (Figs 4b,c and 6a). Moreover, the anti-apoptotic capacity of rapamycin correlated with its ability to reduce p62 levels (Fig. 4b). Finally, the results shown in Figs 2a and 3d demonstrated that the activation of mTORC1 during nutrient restriction activated caspase 3 and caspase 8, but it did not affect caspase 9 activation. Thus, we decided to investigate whether p62 plays a mechanistic role in mTORC1-mediated apoptosis induction. Supporting this hypothesis, we first observed that silencing p62 (using siRNA) was sufficient to prevent the activation of apoptosis mediated by LQ treatment, as determined by caspase 8 and PARP cleavage (Fig. 6a). It is noteworthy that, despite the role assigned to p62 in the activation of mTORC1 (ref. 43), silencing p62 did not affect the LQ-induced activation of mTORC1 (Fig. 6a), which placed p62 downstream of mTORC1 in the glutaminolysis-induced apoptosis. Conversely, the upregulation (exogenous overexpression) of p62 was sufficient to strongly increase cell death and to activate the cleavage of caspase 8, caspase 3 and PARP specifically in cells incubated in the absence of amino acids, but to a much lesser extent in amino acid fed cells (Fig. 6b,c). Confirming the specific role p62 in the activation of caspase 8 and caspase 3, p62 upregulation did not increase the cleavage of caspase 9 (Fig. 6b). Again, the overexpression of p62 did not affect the inactivation of mTORC1 during amino acid starvation (Fig. 6b), confirming that in our conditions p62 operates downstream of mTORC1. In addition, we corroborated that the upregulation of p62 induced its interaction with caspase 8 specifically when cells are incubated in the absence of amino acids. As shown in Fig. 6d, endogenous caspase 8 co-immunoprocipitated with p62-HA only when cells where incubated in the absence of amino acids. These results strongly suggest that the abnormally high levels of p62 during amino acid restriction are sufficient to promote the interaction of p62 with caspase 8 to induce the cleavage of caspase 8. These results sustain a model in which the unbalanced activation of glutaminolysis inhibits autophagy in an mTORC1-dependent manner. Autophagy inhibition induces high levels of p62, which in turn promotes the activation of caspase 8 and apoptosis upon amino acid imbalance (Fig. 6e).

## Discussion

The results presented herein propose a complete molecular mechanism to explain how the activation of glutaminolysis in the absence of other amino acids induces an unbalanced activation of mTORC1, which promotes apoptosis upon amino acid deprivation. This unprecedented function of both glutamine metabolism and mTORC1 (two well-known pro-proliferative inducers) as activators of cell death in tumour cells place both elements with a potential tumour suppressor functionality that could be exploited in therapy. We observed that the addition of glutamine and leucine at similar concentrations than used in a complete culture medium to amino acid-starved cells (what we called 'LQ treatment') promoted the activation of apoptotic cell death through glutaminolysis, as the inhibition of this process abrogated this cell death induction. The unexpected role of glutaminolysis as a cell death inducing mechanism during nutrient restriction also pointed at the importance of nutritional balance in the control of cancer cell viability and the potential use of this metabolic disequilibrium to identify new metabolic addictions. Further sustaining this concept, previous reports showed that increasing intracellular αKG levels induce apoptosis *in vivo*, although no clear mechanism was provided[44,45]. On the other hand, a recent report showed that glutaminolysis is crucial

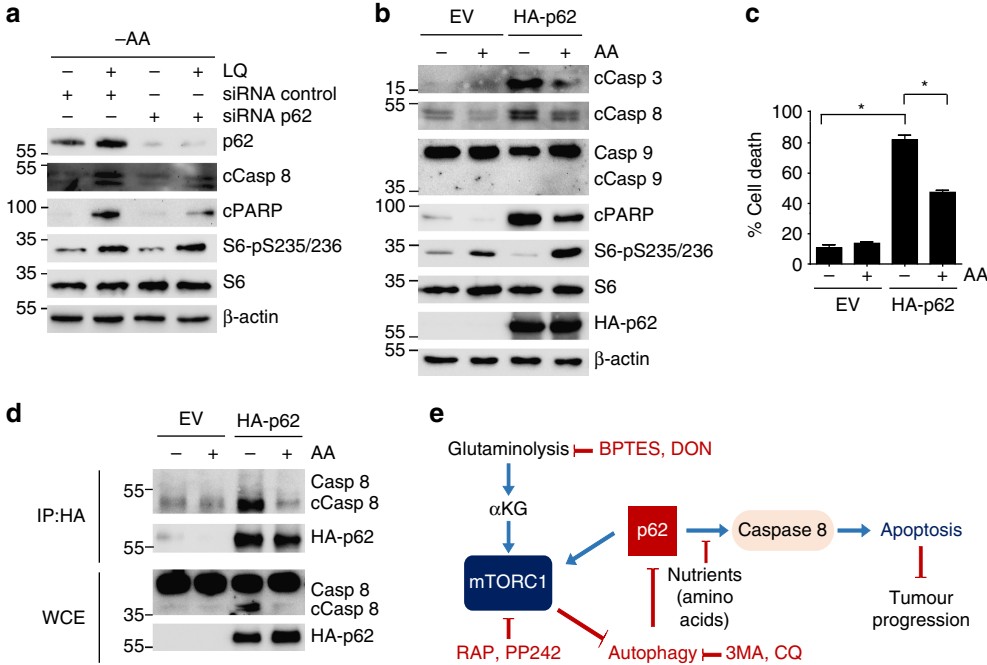

**Figure 6 | p62 interacts with and activates caspase 8 and apoptosis. (a)** U2OS cells were transfected either with a non-targeting siRNA (control) or siRNA against p62. Then cells were treated with LQ for 72 h. The activation of apoptotic markers and mTORC1 downstream targets were assessed by western blot analysis. **(b,c)** U2OS cells were transfected with HA-p62 for 24 h and then incubated in the presence or the absence of all the amino acids for 48 h as indicated. The expression of apoptotic markers and mTORC1 downstream targets were assessed by western blot **(b)**, and cell viability was estimated using trypan blue exclusion assay **(c)**. Graphs show mean values ± s.e.m. (n = 3). *P < 0.05 (Anova *post hoc* Bonferroni). **(d)** U2OS cells were transfected with HA-p62 and the interaction of HA-p62 with endogenous caspase 8 was evaluated by immunoprecipitation upon amino acid starvation ( − AA) or amino acid sufficiency ( + AA). The co-precipitation of bot HA-p62 and caspase 8 was determined by western blot analysis. **(e)** Working model summarizing the results obtained in this work.

for the induction of ferroptosis, a non-apoptotic type of cell death[46].

As we previously described, glutaminolysis activates mTORC1 in a short-term setting[14]. Now, we have corroborated this observation in a long-term setting, showing that the activation of glutaminolysis or its end-up product αKG maintain the activity of mTORC1 for 72 h at least, which correlated with the activation of apoptosis. Strikingly, the inhibition of mTORC1 promoted cell survival upon amino acid starvation. Regarding this observation, it is important to mention here that an abnormally high activity of mTORC1 in nutrient limiting conditions is a stressful situation that many tumour cells (particularly solid tumours) are subjected to, as mTORC1 is aberrantly activated in 80% of human cancer, and the tumour environment is restrictive *per se*[3,8]. According to our results, the addition of rapamycin in these conditions constitutes an opportunity for the cancer cell to resist apoptosis. Indeed, this anti-apoptotic effect of rapamycin might explain, at least in part, the lack of efficacy observed in patients treated with mTORC1 inhibitors: mTORC1 inhibition in patients will indeed restrict tumour growth, but at the same time allows tumour survival and apoptosis resistance, which might lead to an increase in therapy resistance. A number of reports suggest that one of the reasons to explain the limited efficacy of rapamycin to target tumour growth is the specificity of rapamycin to target mTORC1 and not mTORC2 (refs 47–50). As a result, dual inhibitors of both complexes have been developed in the last years. However, we observed that a double inhibitor of mTORC1/mTORC2 (PP242) also promoted cell survival in these conditions, implying that a double mTORC1/mTORC2 inhibition might present similar problems of apoptosis resistance in patients.

In our study, we have dissected the mechanism that promotes cell death by the unbalanced activation of mTORC1 induced by glutaminolysis in the absence of amino acids. Thus, we tested the potential role of UPR, ER stress and autophagy, all processes related with apoptosis and mTORC1 activation[1,2,8,15,16,18,24,29,30]. We observed that none of the tested markers of UPR and ER stress were affected in these conditions. By contrast, the glutaminolysis-mediated activation of mTORC1 inhibited autophagy, a process crucial for the survival of the cells upon nutrient deprivation. Indeed, the inhibition of autophagy was sufficient to induce cell death. We also demonstrated that rapamycin requires autophagy to promote cell survival in glutaminolysis activated cells. Finally, we observed that autophagy-dependent reduction of p62 levels during amino acid withdrawal is a necessary step to prevent apoptosis. Hence, any tested condition that induced high levels of p62 during amino acid restriction (LQ treatment, DMKG treatment, 3MA treatment, ATG5 − / − , p62 overexpression) led to the activation of apoptosis. Therefore, this upregulation of p62 during nutrient restriction seems to be the ultimate mechanism detected by the cell to recognize an anomalous activation of cell growth signalling in restrictive conditions, and that situation prompts the cell to undergo apoptosis. This model (Fig. 6e) was corroborated by the direct interaction of p62 with cleaved caspase 8, a mechanism that has been previously described to activate apoptosis in other stressful circumstances[38–41]. However, the physiological and biochemical control of the interaction between p62 and caspase 8 needs further investigation. Mainly, how amino acid sufficiency prevents this interaction is a question that remains to be answered. Interestingly, the overexpression of p62 in conditions of amino acid sufficiency led to a higher activation of mTORC1, compared to cells expressing normal levels of p62 (Fig. 6b). This corroborates the partial role of p62 in the activation of mTORC1, as described elsewhere[43,51]. This evidence highlights the dual role of p62 either promoting cell growth through mTORC1 pathway

or acting as an apoptotic signal depending on the presence or absence of amino acids.

Altogether, these results points towards autophagy as an 'addiction' in rapamycin-treated tumours, and therefore highlight the potential of autophagy as a therapeutic target to overcome rapamycin-resistant problems in tumour therapy[8,52]. Some clinical trials using the inhibition of both mTORC1 and autophagy have already shown promising results[53,54]. Here, we propose a complete molecular mechanism highlighting the functionality of mTORC1 not only as a major tumour promoter as it has been extensively characterized, but also exhibiting tumour suppressor features during nutrient restrictive conditions. Our results provide with a molecular explanation for the modest results obtained when mTOR inhibitors are used as anti-tumour therapy, as rapamycin treatment promotes survival during nutrient-restricted conditions. Discontinuation of the treatment will be then followed by a relapsed growth of tumour cells that resisted apoptosis induction. In this context, other independent reports support the notion that the inhibition of mTORC1 promotes malignancy in solid tumours. Indeed, Mikaelian *et al.* reported that mTORC1 inhibition promotes the epithelial-mesenchymal transition, increasing the migration of cancer cells[55]. In addition, recently Palm *et al.*[56] observed that the inhibition of mTORC1 increases the use of extracellular sources of nutrients, thus, sustaining the growth of cancer cells exposed to nutrient-restricted conditions. Finally, the results here exposed unveil both the crucial role played by autophagy and p62 in the anti-apoptotic effect of rapamycin and their potential use as therapeutic co-targets in rapamycin therapies.

## Methods

**Reagents and antibodies.** Antibodies against mTOR (#2983, dilution 1:150), S6 (#2217, dilution 1:1,000), phospho-S6 (Ser235/236) (#4856, dilution 1:1,000), S6K (#2708, dilution 1:1,000), phospho-S6K(T389) (#9205, dilution 1:1,000), 4EBP1 (#9452, dilution 1:1,000), phospho-4EBP1(T37/46) (#2855, dilution 1:1,000), AKT (#4691, dilution 1:1,000), phospho-AKT(Ser473) (#4060, dilution 1:1,000), p62 (#5114, dilution 1:1,000), LC3 AB (#12741, dilution 1:1,000), β-actin (#4967, dilution 1:1,000), RAPTOR (#2280, dilution 1:1,000), RICTOR (#2140, dilution 1:1,000), cleaved caspase 3 (#9664, dilution 1:1,000), cleaved PARP (#5625, dilution 1:1,000), Bax (#5023, dilution 1:1,000), caspase 8 (#9746, dilution 1:1,000), caspase 9 (#9508, dilution 1:1,000), ATG5 (#12994, dilution 1:1,000), TNF-α (#3707, dilution 1:1,000), FasL (#4273, dilution 1:1,000), cytochrome c (#4272, dilution 1:1,000), TSC2 (#4308, dilution 1:1,000) and Cox4 (#4850, dilution 1:1,000) were obtained from Cell Signaling Technology. Antibodies against CD63 (SAB4700215, dilution 1:400) and HA (H3663, dilution 1:5,000) were obtained from Sigma. Antibodies against GLS (ab93434, dilution 1:1,000) and Herpud1 (ab155778, dilution 1:1,000) were purchased from Abcam. Antibodies against Fas (sc-715, dilution 1:1,000), BiP (sc-15897, dilution 1:1,000) and Hsp90 (sc-69703, dilution 1:1,000) were obtained from Santa Cruz. Antibody against phospho-EIF2A (Ser52) (44-728G, dilution 1:1,000) was purchased from Thermo Fisher Scientific. Antibody against PDI (ADI-SPA-890, dilution 1:1,000) was obtained from Enzo Life Sciences. The secondary antibodies anti-mouse (#7076, dilution 1:1,000) and anti-rabbit (#7074, dilution 1:1,000) were obtained from Cell Signaling Technology. The apoptotic ligand FasL was kindly provided by Patrick Legembre (INSERM, Rennes, France), while TRAIL mAb (HS501) was obtained from Adipogen. The Permeable αKG (dimethyl-α-ketoglutarate), Diazo-5-oxo-L-norleucine (DON), Bis-2-(5-phenylacetamido-1,2,4-thiadiazol-2-yl)ethyl sulphide (BPTES), Rapamycin (RAP), paraformaldehyde, violet crystal, 3MA, chloroquine (CQ), PP242 were obtained from Sigma. siRNA against GLS1, RAPTOR, RICTOR, p62, ATG5, BAX and non-targeting siRNA control were obtained from Dharmacon. EGFP-LC3 plasmid was a gift from Karla Kirkegaard (Addgene plasmid #11546). HA-p62 plasmid was a gift from Qing Zhong (Addgene plasmid #28027).

**Cell culture.** U2OS, HEK293A, A549 and JURKAT cells were obtained from ATTC. WT and ATG5 − / − MEFs were kindly provided by Patricia Boya (Centro de Investigaciones Biologicas, Madrid, Spain). GFP-LC3 expressing U2OS cells were obtained from Eyal Gottlieb (Cancer research UK, Glasgow, UK). Except for JURKAT (RPMI GIBCO), all the cells lines were grown in DMEM high glucose (4.5 g l$^{-1}$) (GIBCO) supplemented with 10% of fetal bovine serum (Dominique Dutscher), glutamine (2 mM), penicillin (Sigma, 100 U ml$^{-1}$) and streptomycin (Sigma, 100 μg ml$^{-1}$), at 37 °C, 5% $CO_2$ in humidified atmosphere. Mycoplasma contamination check was carried out using the VenorGeM Kit (Minerva Biolabs GmbH, Germany). Standard starvation medium was EBSS (GIBCO) containing

$4.5 \, g \, l^{-1}$ of glucose. The activation of glutaminolysis was performed by adding glutamine (2 mM final concentration) and leucine (0.8 mM final concentration). When indicated, DMKG was added to a final concentration of 0.2–2 mM. The different inhibitors were used concomitantly with the activation of glutaminolysis as follows: DON (40 μM), BPTES (30 μM), rapamycin (100 nM) and PP242 (100 nM).

**Plasmids and siRNA transfections.** The plasmid transfections were carried out using Jetpei (Polyplus Transfection) according to the manufacturer's instructions. Briefly, 70% confluent cells were transfected with 2–3 μg of plasmid. Twenty-four hours later cells were starved in the presence or absence of LQ for 48 h more. siRNA transfections were performed using Interferin@ (Polyplus Transfection) according to the manufacturer's instructions: cells at 50% of confluence were transfected with siRNA (final concentration 10 nM) in complete medium for 48 h and then starved with different treatments for another 72 h.

All siRNAs were obtained from Dharmacon (on-target plus smartpool siRNA). Sequences of the siRNAs were as follows:

Non-targeting control (D-001810-02-05): (1) UGGUUUACAUGUCGACUAA, (2) UGGUUUACAUGUUGUGUGA, (3) UGGUUUACAUGUUUUCUGA, (4) UGGUUUACAUGUUUUCCUA; BAX (L-003308-01-0005): (1) GUGCGGA ACUGAUCAGAA, (2) ACAUGUUUUCUGACGGCAA, (3) CUGAGCAGAU CAUGAAGAC, (4) UGGGCUGGAUCCAAGACCA; ATG5 (L-004374-00-0005): (1) GGCAUUAUCCAAUUGGUUU, (2) GCAGAACCAUACUAUUUGC, (3) UGACAGAUUUGACCAGUUU, (4) ACAAAGAUGUGCUUCGAGA; p62 (L-010230-00-0005): (1) GAACAGAUGGAGUCGGAUA, (2) GCAUUGAAGUU GAUAUCGA, (3) CCACAGGGCUGAAGGAAGC, (4) GGACCCAUCUGUCUU CAAA; Raptor (L-004107-00-0010): (1) UGGCUAGUCUGUUUCGAAA, (2) CACGGAAGAUGUUCGACAA, (3) AGAAGGGCAUUACGAGAUU, (4) UGGAGAAGCGUGUCAGAUA; Rictor (L-016984-00-0010): (1) GACACAA GCACUUGCGAUUA, (2) GAAGAUUUUGAGUCCUA, (3) GCGAGCGUGAUG UAGAAUUA, (4) GGGAAUGAACAACUCCAAAUA; GLS (L-004548-01-0010): (1) CCUGAAGCAGUUCGAAAUA, (2) CUGAAUAUGUGCAUCGAUA, (3) AGAAAGUGGAGAUCGAAAU, (4) GCACAGACAUGGUUGGUAU.

**Immunoblots.** $5 \times 10^6$ JURKAT cells or $2 \times 10^6$ U2OS, A549, HEK293 cells were seeded in 10 cm plates. After the respective treatments cells were washed two times with phosphate-buffered saline (PBS) and lysed with RIPA buffer containing a cocktail of protease inhibitor (P8340 Sigma), inhibitors of phosphatases (P0044 and P5726 Sigma) and PMSF 1 mM. Protein quantification was performed using BCA kit (Pierce). After the electrophoresis, the proteins were transferred to a nitrocellulose membrane (midi kit, Bio-Rad) with Trans-Blot Turbo Transfer System (Bio-Rad). Finally, membranes were imaged using the ChemiDoc MP imager (Bio-Rad). Uncropped Western Blot scan is reported in Supplementary Fig. 6.

**Immunoprecipitation.** After the transfection with p62-HA, the cells were starved for 48 h. After two washes with cold PBS, cells were lysed with lP lysis buffer (40 mM Hepes pH 7.5, 120 mM NaCl, 1 mM EDTA, 0.3% CHAPS, protease inhibitor cocktail P8340 Sigma and 1 mM PMSF). Protein extracts were incubated overnight at 4 °C with anti-HA magnetic beads (Pierce Anti-HA Magnetic Beads, Thermo Fisher #88836). Thereafter beads were washed twice with cold PBS and eluted with Laemmli buffer for immunoblot analysis.

**Cell proliferation and cell viability.** $1.2 \times 10^5$ cells were seeded for all the cell lines (U2OS, A549, HEK293A, JURKAT) and the number of viable cells was determined after 24–144 h, using the TC20 Automated Cell Counter (Bio-Rad) according to the manufacturer's protocol. Briefly, after the respective treatments cells were detached with trypsin/EDTA and 10 μl of the cells suspension were mixed with 10 μl trypan blue 5% solution (Bio-Rad) and analysed with the TC20 cell counter (Bio-rad). To estimate the percentage of cell death, cells were seeded at $1 \times 10^6$ in 6 cm plates and after the treatments (72–144 h), the viability and cell size was assessed with the TC20 cell counter.

**Real-time PCR.** The mRNAs from cells were isolated using Trisol (Invitrogen). One microgram of total mRNA was reverse transcribed using GoScript Reverse Transcription System (Promega). Real-time PCR was performed using SSO Advanced Universal SYBR Green Supermix (Bio-Rad). Specific primers for BAX (forward: CATGTTTTCTGACGGCCAACTTC; reverse: AGGGCCTTGAGCAC CAGTTT, PMM1 (forward: GACAGCTTGACACCATCCA; reverse: CGGCAAA GATCTCAAAGTCGTT) and RPL29 (forward: GGCTATCAAGGCCCTCGT AAA; reverse: CGAGCTTGCGGCTGACA) were purchased from Sigma-Aldrich.

**Subcellular fractionation.** $25 \times 10^6$ cells were seeded in two 25 cm plates for each condition and after the respective treatment the cells were subjected to a subcellular fractionation using the Cell Fractionation Kit (#9038) of Cell Signaling Technology, following the manufacturer's recommendations.

**Flow cytometry.** After treatment, cells were stained with annexin V and propidium iodide (PI) (Annexin V—early apoptosis detection kit, #6592 Cell Signaling Technology) following the manufacturer's instructions. Then, cells were analysed using BDFACS Canto BD-Biosciences flow cytometer. The analysis of the data was performed using the free software Flowing.

**Confocal microscopy.** $1.2 \times 10^5$ cells were grown in coverslips with the respective treatments for 72 h. Thereafter, cells were fixed with 4% paraformaldehyde in PBS during 30 min at room temperature. GFP-LC3 expressing U2OS cell lines were mounted after the fixation with Prolong containing DAPI (Invitrogen). For the co-localization experiments, after the fixation, cells were permeabilized using Triton-X 0.05% during 10 min, and then blocked with BSA 5% in PBS for 30 min. Finally, cells were incubated with the primary antibodies for 1 h at 37 °C. After three washes with PBS, the cover slide was incubated for 1 h at 37 °C with the appropriate secondary antibody (anti-rabbit Alexa488, dilution 1:400 or anti-mouse Alexa555, dilution 1:400, both from Invitrogen). Finally, coverslips were mounted with Prolong (invitrogen). Samples were imaged with a Leica Confocal microscope.

**Clonogenic assay.** Cell were starved in EBSS (glucose $4.5 \, g \, l^{-1}$) with or without leucine/glutamine or DMKG during 72 h for U2OS and A549, and during 144 h for HEK293A. Similarly, U2OS cells were starved for amino acids, treated with LQ, RAP (100 nM) and/or 3MA (5 mM) as indicated for 72 h. After the treatment, $1.5 \times 10^3$ cells (U2OS, A549) or $3 \times 10^4$ (HEK293A) were seeded in a 3 cm plate containing complete media. After 14 days cells were fixed with paraformaldehyde 4% in PBS (30 min) and stained with violet crystal 5% for 15 min. Then, the plates were washed with water and imaged using ChemiDoc MP Imager (Bio Rad).

**Transmission electron microscopy.** After the respective treatment, cells were fixed for 1 h at 4 °C in 4% paraformaldehyde in PBS, washed and fixed again 1 h at room temperature in aqueous 2% osmium tetroxide in 0.2 M sodium cacodylate (pH 7.4). Dehydration was performed with ethanol (50%, 70%, 95% and absolute ethanol). Thereafter, the samples were embedded in Epon/Ethanol and evaporated overnight at room temperature. The samples were processed for ultra-microtomy according to standard procedures. Finally sample imaging was performed using a Hitachi H7650 microscope operated at $-80 \, KV$ with a camera Gatan—11 MPx.

Autophagosome formation was quantified counting the number of autophagy-related vesicles per area in several images for each condition and the data are represented as the average number of vesicles per $\mu m^2$.

**Statistics.** The results are expressed as a mean ± s.e.m. of at least three independent experiments. One-way ANOVA followed by Bonferroni's comparison as a *post hoc* test were used to evaluate the statistical difference of the results. Statistical significance was estimated when $P < 0.05$.

**Data availability.** The authors declare that all the data supporting the findings of this study are available within the article and its Supplementary Information files and from the corresponding author on reasonable request.

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

## Acknowledgements

This work was supported by funds from the following institutions: Institut National de la Santé et de la Recherche Médicale—INSERM, Fondation pour la Recherche Médicale, the Conseil Régional d'Aquitaine, Fondation ARC pour la Recherche sur le Cancer, SIRIC-BRIO, Institut Européen de Chimie et Biologie. We thank Vincent Pitard and Santiago Gonzalez (Flow Cytometry Platform, Université de Bordeaux, France) for technical assistance in flow cytometry experiments. MEFs WT and ATG5 − / − were kindly provided by Prof. Patricia Boya (Centro de Investigaciones Biologicas, Madrid, Spain). FasL was a gift from Patrick Legembre (INSERM, Rennes, France). EGFP-LC3 plasmid was a gift from Karla Kirkegaard. HA-p62 plasmid was a gift from Qing Zhong.

## Author contributions

V.H.V. and R.V.D. conceived the project. V.H.V., M.B., P.V., M.P., and R.V.D. designed experiments. V.H.V., T.L.N., S.T., V.D., M.B., C.B., B.S. and M.P. performed experiments and analysed data. R.V.D. and P.S. secured funding. V.H.V. and R.V.D. wrote the manuscript. All the authors read and approved the manuscript.

## Additional information

**Competing financial interests:** The authors declare no competing financial interests.

