## [Peer Review File · Nature Communications]

Reviewers' comments:

Reviewer #1 -Expert in Autophagy and apoptosis
(Remarks to the Author):

The manuscript by Villar and colleagues describes a novel role for mTORC1 in the suppression of cell death caused by hyper-glutaminolysis. Although the physiological relevance of exposure of cells to Glutamine and Leucine in the absence of other amino acids is somewhat abstract, the study does provide new mechanistic insights into starvation responses and mTORC1 function that may be relevant for certain therapeutic situations. I feel that the study would be of interest to those working in glutaminolysis and/or mTORC1, but I consider the manuscript has a number of deficiencies in current form that preclude publication at this time. The points listed below are not in any particular order regarding priority or importance.

1) In Figure 1, the data using the GLS inhibitor would be greatly enhanced by studies involving RNAi-mediated knockdown of GLS.

2) On p. 6, the authors refer to data in Figure 2 as follows, '.....DMKG treatment increased the levels of cleaved caspase 3, PARP and Bax in U2OS, A549, JURKAT and HEK293A cells'. In fact, the levels of Bax are only assessed in U2OS and HEK293A. The text should be revised accordingly. A similar situation also exists on p. 9 when the authors refer to data in Figure 3 and accompanying Supplementary data. Again, the text should be revised accordingly.

3) On p. 8, the authors state, 'These results confirmed that long-term glutaminolysis, even in the absence of other amino acids, is....'. This statement is not strictly correct as the experiments were also done in the presence of Leucine. The text should be revised accordingly.

4) On p. 8, the authors refer to CD63 as a lysosome marker. CD63 is also well characterized as a marker of late endosomes and the authors should consider this in fact when interpreting their data.

5) Since the authors show that the cell death response involves caspase 8 and mTORC1 regulates translation, have the authors considered that mTORC1 inhibition might restrict translation of TNF, FasL or TRAIL as a mechanism to explain cell death via caspase 8?

6) In reference to Supp Fig 3H, the authors state that because cycloheximide did not prevent death then the UPR and ER stress are not involved in glutaminolysis/mTORC1-mediated apoptosis. The authors cannot make this claim since treatment with CHX alone induces PARP cleavage.

7) Can the authors provide any insights into how Bax is up-regulated in response to LQ? Is this transcriptional? If so, what transcription factors might be involved?
In addition, the authors should explore the importance of Bax in LQ-induced cell death by either RNAi-mediated knockdown or CRSIPR/Cas9-mediated disruption. This is an important experiment in light of the fact that the authors conclude that the cell death does not involve caspase 9. Further data in these areas would significantly strengthen the mechanism.

8) In Figure 3D, the authors consider that cell death induced by LQ might involve up-regulation of pro-apoptotic members of the Bcl-2. The authors should also explore whether anti-apoptotic members of this family are down-regulated.

9) In Figure 4A, the authors analyze cells for LC3-positive puncta at a time point where they have shown 60% of the cell population is dying (see Fig 1C). Since the images shown in 4A appear to be relatively healthy cells, can comparisons be made here regarding the induction of autophagy and cell death. In other words what is the situation regarding LC3-positive puncta in the majority of the population that is undergoing cell death? This comment also applies to Figure 5A. Also, with

respect to 5A, the authors must indicate how long the cells have been incubated in 3-MA.

10) Figure 4B requires a rapamycin alone control. In addition, to correctly make statements about autophagic flux, the authors need to repeat the treatment in Figures 4B and 4C in either the absence or presence of chloroquine to trap the formation of autophagosomes.

11) For completeness, Figure 5G requires an LQ treatment +/- rapamycin in both cell lines.

12) In the IP shown in Figure 6D, the authors need to show total as well as cleaved caspase 8.

13) In Figures 4H, 5G and S5B, the authors refer to a band as ATG5. It is assumed this band is actually the ATG5-ATG12 complex. If so, it is important that the authors change the annotation of these figures. Also, to clearly show whether this is ATG5 or ATG5-ATG12, molecular weight markers on these westerns would be useful.

Reviewer #2 -Expert in mTOR and cancer
(Remarks to the Author):

The manuscript by Villar et al. provides evidence that, during amino acid starvation, addition of leucine and glutamine is sufficient to activate mTORC1 through the glutaminolysis pathway and sensitizes cells to cell death. Since mTOR inhibition by rapamycin rescues cell viability in these conditions, the authors propose that mTOR may act as a tumour suppressor favouring cancer cell death in conditions on nutrient limitation.

The data in the cell lines are clear and well presented. However, at this stage the authors overstate the novelty and implications of their conclusions. That mTOR overactivation increases cell susceptibility to apoptosis and senescence is known since a long time. There is a huge literature, mainly using TSC1/2 mutant cells, demonstrating this point which then led to the idea of testing combinations of mTOR inhibitors and autophagy inhibitors.

Major issues:

- 1) It is unclear in what pathophysiological conditions these observations may be relevant. The authors should consider developing an in vivo cancer system to test their hypothesis. Do they expect that GLS1 knockdown would promote cancer progression in a context of nutrient limitation and unbalance?
- 2) What cancer genotype would reveal these effects of mTOR on the inhibition of cancer cell survival? The definition of mTOR as a tumour suppressor is very strong and precise. It should be validated by appropriate assays using genetic gain- and loss-of-function of mTOR pathway. The in vitro clonogenic assays reported in this study are far from being the adequate method.
- 3) A long term 3MA treatment is a rather unspecific method to inhibit autophagy. Knock-out and knock-down cell lines for autophagy genes should be used with the whole battery of treatments, including LQ stimulation and rapamycin.

Reviewer #3 -Expert in mTOR inhibitors and cancer
(Remarks to the Author):

Duran and colleagues have investigated a potential tumor-suppressing role of mTORC1 under conditions of conflicting nutrient signals. Specifically they found that glutaminolysis-dependent activation of mTORC1 in otherwise amino-acid-starved cells triggers apoptosis; and, that inhibition of mTORC1 with rapamycin rescues the ensuing cell death. They subsequently demonstrated that it is the rapamycin-triggered upregulation of autophagy that prevents apoptosis via a reduction of p62 protein levels and a subsequent reduction of Caspase 8 and 3 activation. From their results, the authors suggest a potential explanation for the lack of results of mTORC1 inhibitors in many oncology trials.

The manuscript generally reads well and the conclusions are supported by the data presented. Not all aspects of this story are novel but the authors do a very good job of synthesizing a coherent, unifying and exciting message from their data combined with the mTOR literature. Pending minor comments listed below, I find this manuscript suitable for publication in Nat. Comm.

Specific comments:

1. Figures 4H and 5G: The text referring to these figures indicates that cell survival was assessed. However, the figures themselves are western blots of apoptotic markers. The presented data is important and should be kept. But the cell survival data mentioned in the text should additionally be presented.
2. I had difficulties understanding the significance of the EM data in figure 4D. To what are the arrow heads pointing? Can this data be somehow quantified? Can some landmark features be indicated (lysosome for example)?
3. The wording of the abstract is a bit awkward, particularly the sentence starting on line 32. In general, aa starvation when LQ are present in the medium is a bit confusion. A better distinction between aa starvation and aa imbalance should be made to help guide the reader.
4. The interpretation of the data in Fig 6 B-C is over-interpreted. On line 349 of the manuscript "...cells incubated in the absence of amino acids, but it did not affect apoptosis in amino acid fed cells (Figure 6B-C)." "it did not affect apoptosis" should be replaced with "to a much lesser extent".

Point-by-point answers to the reviewers' comments

Reviewer #1 -Expert in Autophagy and apoptosis

(Remarks to the Author):

The manuscript by Villar and colleagues describes a novel role for mTORC1 in the suppression of cell death caused by hyper-glutaminolysis. Although the physiological relevance of exposure of cells to Glutamine and Leucine in the absence of other amino acids is somewhat abstract, the study does provide new mechanistic insights into starvation responses and mTORC1 function that may be relevant for certain therapeutic situations.

We thank Reviewer #1 for the very helpful suggestions and comments. We understand that the approach that we have followed (addition of leucine and glutamine in the absence of other amino acids) might exceed a particular physiological condition. However, the hyper-activation of both glutaminolysis and mTORC1 signaling even in conditions of nutrient limitations are two common features of solid tumors, as discussed in the manuscript. In this context, our results provide helpful insight to understand the consequences of nutritional imbalance at the level of tumor cell viability and, particularly, the use of mTORC1 inhibitors in the clinic, highlighting the physiological relevance and the involvement in cancer therapy of our results.

I feel that the study would be of interest to those working in glutaminolysis and/or mTORC1, but I consider the manuscript has a number of deficiencies in current form that preclude publication at this time. The points listed below are not in any particular order regarding priority or importance.

1) In Figure 1, the data using the GLS inhibitor would be greatly enhanced by studies involving RNAi-mediated knockdown of GLS.

As suggested, in the new version of the manuscript we included studies involving RNAi-mediated knockdown of GLS in Figure 1, and also in Figure 2 and Supplementary Figure S4. As shown in Figure 1F-G, Supplementary Figure S1G and Figure 2H, the efficient silencing of GLS prevented LQ-induced apoptosis both in U2OS and in HEK293 cells, in agreement with the results obtained using DON and BPTES. We extended the analysis using siRNA against GLS to confirm that the knockdown of GLS was also sufficient to activate autophagy in LQ-treated cells (Supplementary Figure S4G), again confirming the results previously obtained using DON and BPTES.

2) On p. 6, the authors refer to data in Figure 2 as follows, '.....DMKG treatment increased the levels of cleaved caspase 3, PARP and Bax in U2OS, A549, JURKAT and HEK293A cells'. In fact, the levels of Bax are only assessed in U2OS and HEK293A. The text should be revised accordingly. A similar

situation also exists on p. 9 when the authors refer to data in Figure 3 and accompanying Supplementary data. Again, the text should be revised accordingly.

We apologise for this imprecision. In the new version of the manuscript we assessed BAX levels by western blot also in A549 and JURKAT cells (Supplementary Figure S2A,B). Regarding Figure 3, we have amended the text accordingly.

3) On p. 8, the authors state, 'These results confirmed that long-term glutaminolysis, even in the absence of other amino acids, is...'. This statement is not strictly correct as the experiments were also done in the presence of Leucine. The text should be revised accordingly.

We apologise for this imprecision. As requested, in the new version of the manuscript we amended the text in the following manner: 'These results confirmed that long-term activation of glutaminolysis adding glutamine and leucine, even in the absence of other amino acids, is...'

4) On p. 8, the authors refer to CD63 as a lysosome marker. CD63 is also well characterized as a marker of late endosomes and the authors should consider this in fact when interpreting their data.

We apologise for this imprecision. We have corrected the text accordingly, specifying that CD63 is a marker of late endosomes and lysosomes. However, this difference does not impact the interpretation of our data, as mTORC1 has been previously shown very extensively by ourselves and other authors to translocate specifically to the lysosome (see Bar-Peled & Sabatini Trends Cell Biol. 2014 Jul;24(7):400-6 for a review).

5) Since the authors show that the cell death response involves caspase 8 and mTORC1 regulates translation, have the authors considered that mTORC1 inhibition might restrict translation of TNF, FasL or TRAIL as a mechanism to explain cell death via caspase 8?

We thank the reviewer for this suggestion. As suggested, we have now investigated if mTORC1 inhibition using rapamycin has an impact on protein levels of FasL, TNFa or TRAIL. As shown in Supplementary Figure S3J, we clearly observed that rapamycin treatment did not restrict protein expression of any of those ligands, as determined by protein levels upon rapamycin addition. Indeed, we did not detect any protein levels of either FasL or TNFa upon LQ addition in U2OS cells, and rapamycin did not affect this result. On the other hand, we detect basal levels of TRAIL, but no increase/decrease was observed upon LQ or rapamycin treatment. The protein levels of the death receptor Fas were not affected either. Thus, we concluded that neither the glutaminolysis-dependent induction of cell death, nor the protective effect of rapamycin involved the translational control of FasL, TNFa or TRAIL as a mechanism to activate caspase 8 cleavage.

To further discard a potential role of FasL or TRAIL in the protective effect of rapamycin, we also observed that rapamycin treatment did not protect from FasL or TRAIL-induced cell death in U2OS, and JURKAT cells. As shown in Supplementary Figure S3K-N, TRAIL induced cell death in JURKAT cells but not in U2OS cells, while FasL treatment induced cell death in both cell lines (as determined

by AnnexinV/PI staining and by cleaved caspase 3/8). However, we did not observed any protective effect in cells co-treated with rapamycin. Thus, rapamycin was unable to protect from FasL/TRAIL induced cell death.

Altogether, those results strongly suggest that the capacity of mTORC1 inhibition to prevent glutaminolysis-dependent cell death does not involve translational modifications of TNF α , FasL or TRAIL, as suggested by the reviewer.

6) In reference to Supp Fig 3H, the authors state that because cycloheximide did not prevent death then the UPR and ER stress are not involved in glutaminolysis/mTORC1-mediated apoptosis. The authors cannot make this claim since treatment with CHX alone induces PARP cleavage.

We agree with this comment made by this reviewer. In the new version of the manuscript, we decided to eliminate the results involving CHX treatment, as the interpretation of these data might be confusing.

7) Can the authors provide any insights into how Bax is up-regulated in response to LQ? Is this transcriptional? If so, what transcription factors might be involved?

We thank this reviewer for this important observation. As requested, now we have investigated if LQ treatment modified the transcriptional levels of BAX. For this purpose, we determined by qPCR if the previously observed changes in the protein levels of BAX upon LQ addition correlated with changes in BAX mRNA levels. As shown in Supplementary Figure S2L, LQ treatment did not change the levels of BAX mRNA, thus discarding that the changes observed in BAX can be explained by a transcriptional regulation of BAX upon LQ treatment. In agreement with this observation, we also observed that p53 signaling (a well-known up-regulator of BAX transcription) was not increased either upon LQ treatment (Supplementary Figure S2M). We conclude that the regulation of BAX by LQ follows a non-canonical mechanism. We found this a very intriguing and interesting point in our investigations, and we intend to extend our work to elucidate the exact mechanism of BAX upregulation in LQ treated cells in a future work.

In addition, the authors should explore the importance of Bax in LQ-induced cell death by either RNAi-mediated knockdown or CRSIPR/Cas9-mediated disruption. This is an important experiment in light of the fact that the authors conclude that the cell death does not involve caspase 9. Further data in these areas would significantly strengthen the mechanism.

As requested, we have now investigated the importance of BAX in LQ-induced cell death using RNAi-mediated knockdown of BAX. As shown in Supplementary Figure S2J, the efficient knockdown of BAX strongly prevented LQ-induced cell death. Furthermore, BAX silencing drastically reduced caspase 3/8 and PARP cleavage in LQ-treated cells, suggesting an active role of BAX in LQ-mediated apoptosis (Supplementary Figure S2K). However, canonical BAX-dependent cell death induction was not observed, as we did not observe Caspase 9 cleavage or cytochrome c

release upon LQ treatment (Figure 2A and Supplementary Figure S2N). The lack of capacity of BAX to induce Caspase 9 cleavage and cytochrome c release could be explained by the presence of anti-apoptotic factors, such as Bcl-XL or MCL1, that might be counteracting the action of BAX upon LQ-treatment (Figure 2A). Thus, our results support a model in which BAX is necessary for the activation of LQ-induced apoptosis, but following a mechanism that does not involve the canonical activation of caspase 9 and cytochrome c release.

8) In Figure 3D, the authors consider that cell death induced by LQ might involve up-regulation of pro-apoptotic members of the Bcl-2. The authors should also explore whether anti-apoptotic members of this family are down-regulated.

As requested, in the new version of the manuscript we investigated if LQ treatment downregulated anti-apoptotic members of the Bcl2 family, such as Bcl-XL and MCL1. Contrary to the reviewer's suggestion, none of these anti-apoptotic factors were downregulated by LQ treatment (Figure 2A). In the case of MCL1, we even observed an increase rather than a decrease in the protein levels. This result suggested that those anti-apoptotic factors were not involved in the induction of cell death by LQ treatment. Indeed, as discussed in the previous point, the presence of Bcl-XL and MCL1 upon LQ-treatment might counteract the role of BAX, explaining the lack of cleaved caspase 9 and cytochrome c release even when BAX levels are elevated.

9) In Figure 4A, the authors analyze cells for LC3-positive puncta at a time point where they have shown 60% of the cell population is dying (see Fig 1C). Since the images shown in 4A appear to be relatively healthy cells, can comparisons be made here regarding the induction of autophagy and cell death. In other words what is the situation regarding LC3-positive puncta in the majority of the population that is undergoing cell death? This comment also applies to Figure 5A. Also, with respect to 5A, the authors must indicate how long the cells have been incubated in 3-MA.

We thank this reviewer for this observation. We agree that autophagy was assessed in surviving cells, as dead cells escape to the fluorescence analysis by microscopy. To overcome this technical problem, we decided to analyse GFP-LC3 puncta also at earlier stages (8h), before cell death was completely triggered. Using this alternative approach, we confirmed that LQ and DMKG treatments strongly reduced the number of GFP-LC3 puncta preceding cell death induction, and this effect was prevented by rapamycin treatment (Supplementary Figure S4A-B).

Regarding 3-MA treatment, as now indicated in figure legend, the treatment was done for 72h.

10) Figure 4B requires a rapamycin alone control. In addition, to correctly make statements about autophagic flux, the authors need to repeat the treatment in Figures 4B and 4C in either the absence or presence of chloroquine to trap the formation of autophagosomes.

As requested, in the new version of the manuscript we included a rapamycin alone control in Figure 4B. Also, we have now analysed the response of autophagy to LQ and DMKG treatments in

the presence of chloroquine, as suggested by this reviewer. As shown in Supplementary Figure S4A-D, LQ and DMKG treatments were able to reduce LC3-II levels, to reduce GFP-LC3 puncta, and to induce p62 also in the presence chloroquine. This result confirmed that glutaminolysis indeed reduced the autophagic flux in cancer cells.

11) For completeness, Figure 5G requires an LQ treatment +/- rapamycin in both cell lines.

As requested, we have also investigated cell death induction in ATG5^{-/-} MEFs upon LQ treatment both in the presence or the absence of rapamycin (Supplementary Figure S5D-E). However, as ATG5^{-/-} cells are very sensitive to amino acid withdrawal, we did not really observe an increase in cleaved caspase or cleaved PARP levels upon LQ addition, as the untreated control already showed high levels of both apoptotic markers. For this reason, we decided to maintain the original Figure 5G, adding the new panel as a supplementary figure (Supplementary Figure S5D-F).

12) In the IP shown in Figure 6D, the authors need to show total as well as cleaved caspase 8.

Both total and cleaved caspase 8 are now shown in Figure 6D.

13) In Figures 4H, 5G and S5B, the authors refer to a band as ATG5. It is assumed this band is actually the ATG5-ATG12 complex. If so, it is important that the authors change the annotation of these figures. Also, to clearly show whether this is ATG5 or ATG5-ATG12, molecular weight markers on these westerns would be useful.

We thank the reviewer for this observation. As suggested, we have modified the annotation of those panels, substituting "ATG5" by "ATG5-ATG12". For reasons of uniformity along the manuscript, we decided not to include the molecular weights for those particular panels. However, we place here a copy of these blots, including molecular weights, for referee #1.

Figure 4I

Figure 5H

Sup Figure S5B

Reviewer #2 -Expert in mTOR and cancer

(Remarks to the Author):

The manuscript by Villar et al. provides evidence that, during amino acid starvation, addition of leucine and glutamine is sufficient to activate mTORC1 through the glutaminolysis pathway and sensitizes cells to cell death. Since mTOR inhibition by rapamycin rescues cell viability in these conditions, the authors propose that mTOR may act as a tumour suppressor favouring cancer cell death in conditions on nutrient limitation.

The data in the cell lines are clear and well presented.

We thank Reviewer #2 for this very kind comment. We also wanted to thank this reviewer for the very useful suggestions and comments.

However, at this stage the authors overstate the novelty and implications of their conclusions. That mTOR overactivation increases cell susceptibility to apoptosis and senescence is known since a long time. There is a huge literature, mainly using TSC1/2 mutant cells, demonstrating this point which then led to the idea of testing combinations of mTOR inhibitors and autophagy inhibitors.

We apologise if we were not clear on this point. In the first version of our manuscript, we decided not to use cellular models of mTOR overactivation (such as TSC KO MEFs), because our results focused on the capacity of unbalanced glutaminolysis to induce cell death in cancer cells, and its physiological relevance beyond mTORC1 overactivation. However, following the suggestions of Reviewer #2, we have now investigated the role of glutaminolysis-induced cell death in TSC2-/- MEFs (see our answer to point #2). As indicated by Reviewer #2, TSC2-/- MEFs are known to present an increased susceptibility to cell death in response to different metabolic stresses, such as glucose deprivation, serum deprivation or hypoxia. However, the mechanism underlying this increased sensitivity to apoptosis in TSC2-/- MEFs is independent of autophagy, rather involving energetic stress and ER stress (Choo AY et al., Mol Cell. 2010, 38:487-99; Young et al., Genes Dev. 2013, 27:1115-31). Thus, it seems that the mechanism of TSC2-/- sensitivity differs from the mechanism of glutamine-induced apoptosis (which we clearly showed to be dependent on autophagy). Thus, we believe that our observations add a novel mechanism of interaction between mTOR and apoptosis induction that is not explained by the phenotype of TSC-null cells.

We agree with Reviewer #2 that the idea of a combined inhibition of both mTOR signaling and autophagy has been proposed before, and indeed the anti-tumor effects of this combined therapy is currently being investigated in clinical trials. However, our results revealed a new direct mechanistic link connecting the autophagic cargo protein p62 with mTORC1-induced apoptosis. These results have important consequences at the level of designing more specific strategies, as the inhibitors of autophagy that are currently considered present serious problems of specificity. Besides, the potential synergism between mTOR inhibition and autophagy inhibition in the context of nutrient imbalance has not been shown before.

Major issues:

1) It is unclear in what pathophysiological conditions these observations may be relevant. The authors should consider developing an in vivo cancer system to test their hypothesis.

The capacity of unbalanced high levels of alpha-ketoglutarate (the product of glutaminolysis) to induce apoptosis in vivo in cancer cells has been already observed, although no mechanism was provided till now. Indeed, Tennant and Gottlieb (Oncogene 2009, 28:4009-21 and J Mol Med 2010, 88:839-49) showed that alpha-ketoglutarate treatment induced apoptotic cell death in vivo. While these authors indicate that the alpha-ketoglutarate-mediated apoptosis induction followed a HIF-independent mechanism, they failed to propose and validate an actual mechanism of interaction. Our results now provide with a complete explanation for the results of Tennant and Gottlieb, which in turn confirmed the in vivo relevance of our observations. We included a paragraph in the Discussion section of the manuscript regarding this point.

Do they expect that GLS1 knockdown would promote cancer progression in a context of nutrient limitation and unbalance?

We do not expect GLS1 knockdown to promote cancer progression, as we do not claim that GLS1 is a tumor suppressor. While we showed that GLS1 silencing (or inhibition) reduced glutaminolysis-mediated apoptosis, it is known that GLS1 inhibition prevents tumor progression, as glutamine metabolism is essential for tumor growth. Indeed, silencing of GLS1 in LQ-treated cells prevented cell death (Figure 1F and Figure 2H), but did not allow cell proliferation, similar to amino acid-starved cells. Perhaps we should clarify that we do not claim that the axis glutaminolysis/mTORC1 is a tumor suppressor mechanisms. Indeed, in most of the situations it is a tumor promoter mechanism. We claim that in conditions of nutrient restriction (and the solid tumor environment is restrictive per se), the axis glutaminolysis/mTORC1 has tumor suppressor features, which have to be considered when inhibitors of this pathway are used for cancer therapy.

2) What cancer genotype would reveal these effects of mTOR on the inhibition of cancer cell survival?

In order to identify a cancer genotype that might be relevant for the effects of mTORC1 inhibition on cell survival, we investigated if a TSC KO genotype presented an increased susceptibility to LQ-induced apoptosis. Our results clearly showed that TSC2^{-/-} cells are sensitive to LQ-induced cell death, and indeed rapamycin treatment showed a clear protection against apoptosis (Supplementary Figure S3H). Similarly, we also tested T-cell lymphoblastic leukemic cells with and PTEN negative background (JURKAT cells). Similar to TSC, PTEN ablation also renders mTORC1 hyperactivation. As we show in Supplementary Figure S1A-C and Supplementary Figure S2B, JURKAT (PTEN-null) cells are also sensitive to LQ-induced cell death, and again rapamycin treatment rescued cell survival in these cells. These results suggest that a genetic background of

mTORC1 hyperactivation (such as the case of TSC-null and PTEN-null cells) are relevant genotypes to consider the adverse cell survival effects of rapamycin as an anti-cancer treatment.

The definition of mTOR as a tumour suppressor is very strong and precise. It should be validated by appropriate assays using genetic gain- and loss-of-function of mTOR pathway. The in vitro clonogenic assays reported in this study are far from being the adequate method.

Regarding the definition of mTOR as a tumor suppressor, as we stated above, we want to emphasize that we do not claim that mTORC1 is a tumor suppressor, but rather it has tumor suppressor features under restrictive conditions. The tumor promoter functions of mTOR are very well documented, and we do not argue against that. We truly apologise if we were not clear on this aspect. We modified the Discussion section to avoid confusion on this aspect.

3) A long term 3MA treatment is a rather unspecific method to inhibit autophagy. Knock-out and knock-down cell lines for autophagy genes should be used with the whole battery of treatments, including LQ stimulation and rapamycin.

We agree that 3MA treatment is a rather unspecific method to inhibit autophagy. Thus, as requested, we now included ATG5 knock-out and knock-down experiments. As shown in Figure 4H-I, Figure 5G-H and Supplementary Figure S5C-E, ATG5 ablation or ATG5 silencing was sufficient to restore cell death in rapamycin-treated cells, either in the presence or the absence of LQ. Thus, the results obtained using 3MA were confirmed using a more specific approach to inhibit autophagy.

Reviewer #3 -Expert in mTOR inhibitors and cancer

(Remarks to the Author):

Duran and colleagues have investigated a potential tumor-suppressing role of mTORC1 under conditions of conflicting nutrient signals. Specifically they found that glutaminolysis-dependent activation of mTORC1 in otherwise amino-acid-starved cells triggers apoptosis; and, that inhibition of mTORC1 with rapamycin rescues the ensuing cell death. They subsequently demonstrated that it is the rapamycin-triggered upregulation of autophagy that prevents apoptosis via a reduction of p62 protein levels and a subsequent reduction of Caspase 8 and 3 activation. From their results, the authors suggest a potential explanation for the lackluster results of mTORC1 inhibitors in many oncology trials.

The manuscript generally reads well and the conclusions are supported by the data presented. Not all aspects of this story are novel but the authors do a very good job of synthesizing a coherent,

unifying and exciting message from their data combined with the mTOR literature. Pending minor comments listed below, I find this manuscript suitable for publication in Nat. Comm.

We truly thank Reviewer #3 for these very nice comments and fruitful suggestions.

Specific comments:

1. Figures 4H and 5G: The text referring to these figures indicates that cell survival was assessed. However, the figures themselves are western blots of apoptotic markers. The presented data is important and should be kept. But the cell survival data mentioned in the text should additionally be presented.

We apologise for this imprecision. As requested, in the new version of the manuscript we included cell death analysis accompanying the western blot of apoptotic markers (Figure 4H-I and Figure 5G-H).

2. I had difficulties understanding the significance of the EM data in figure 4D. To what are the arrow heads pointing? Can this data be somehow quantified? Can some landmark features be indicated (lysosome for example)?

We apologise if the EM results were not clear. In the new version of the manuscript, we tried to implement the clarity of these images, by clearly indicating the autophagosome structures using arrows. We also quantified these images, by estimating the number of autophagosome vesicles observed per area (Figure 4D).

3. The wording of the abstract is a bit awkward, particularly the sentence starting on line 32. In general, aa starvation when LQ are present in the medium is a bit confusion. A better distinction between aa starvation and aa imbalance should be made to help guide the reader.

As suggested we modified the abstract, to clearly distinguish between aa starvation and aa imbalance.

4. The interpretation of the data in Fig 6 B-C is over-interpreted. On line 349 of the manuscript "...cells incubated in the absence of amino acids, but it did not affect apoptosis in amino acid fed cells (Figure 6B-C)." "it did not affect apoptosis" should be replaced with "to a much lesser extent".

We apologise for this imprecision. As suggested, we replaced the indicated sentence, to avoid over-interpretation of our results.

REVIEWERS' COMMENTS:

Reviewer #1 (Remarks to the Author):

The authors have addressed all my concerns and the manuscript is now much improved. I now fully support publication.

Reviewer #3 (Remarks to the Author):

In this revised version my prior criticisms have been appropriately addressed. The comments of the other reviewers seem also to have been adequately addressed. I think this manuscript presents a robust and coherent piece of work that will be of broad interest and therefore be accepted for publication.